



# Coincident In-situ and Triple-Frequency Radar Airborne Observations in the Arctic

Cuong M. Nguyen[1], Mengistu Wolde[1], Alessandro Battaglia[2,3,4], Leonid Nichman[1], Natalia Bliankinshtein[1], Samuel Haimov[5], Kenny Bala[1] and Dirk Schuettemeyer[6]

[1]Flight Research Laboratory, National Research Council Canada, Ottawa, K1A 0R6, Canada
[2]DIATI, Politecnico di Torino, Torino, Italy
[3]Earth Observation Science, Department of Physics and Astronomy, University of Leicester, Leicester, United Kingdom
[4]National Centre for Earth Observation, University of Leicester, Leicester, United Kingdom
[5]University of Wyoming, Laramie, USA
[6]European Space Agency, Noordwijk, NL

*Correspondence to*: Cuong M. Nguyen (Cuong.Nguyen@nrc-cnrc.gc.ca)

**Abstract.** The dataset collected during the Radar Snow Experiment (RadSnowExp) presents the first-ever triple-frequency radar reflectivities combined with almost perfectly co-located and coincident airborne in situ microphysics probes on board the National Research Council Canada (NRC) Convair-580 aircraft. Over 12 hours of flight data in mixed phased and glaciated clouds with more than 3.4 hours in non-Rayleigh regions for at least one of the radar frequencies provide a unique opportunity for studying the relationship between cloud microphysical properties and radar triple-frequency signals. The in situ particle imagery data for this study include imagery from the CPI probe, which provides high resolution particle imagery and allow accurate identification of particle types including level of riming within the DFR plane. The airborne triple-frequency radar data are analysed and grouped based on the dominant particle compositions and microphysical processes (level of aggregation and riming). The results from this study are consistent with the main findings of previous modelling studies with specific regions of the dual-frequency ratio (DFR) plane associated with unique scattering properties of different ice habits, especially in clouds where radar signal is dominated by large aggregates.. Moreover, the analysis shows that the close relationships between the triple-frequency signatures and particles' bulk density, level of riming and aggregations and characteristic size of the particle size distribution (PSD)

## 1 Introduction

There are currently two spaceborne atmospheric radars in operation: the GPM-DPR and the CloudSat CPR whose missions have been foundational for characterizing the evolving nature of clouds and precipitation on Earth over the last decade. The Cloud Profiling Radar (CPR) on board CloudSat is a 94 GHz nadir-looking radar (Stephens et al., 2008), unique in its ability to sense condensed cloud particles whilst coincidentally detecting precipitation. While the CPR was not specifically designed for rain retrieval, its data have shown a great potential also for rain estimation and snowfall in particular, providing vertical profiles of snowfall rate along with snow size distribution parameters and snow water content. The joint NASA/JAXA GPM mission (Hou et al., 2014), launched at the end of February 2014, aims at providing global measurements of precipitation with a higher accuracy and a wider coverage in latitudinal span (65º) than those obtained by the TRMM mission (Iguchi et al., 2000; Nesbitt and Anders, 2009). The GPM carries a Dual-Frequency Precipitation Radar (DPR) system including a Ka-band (35.5 GHz) radar and a Ku-band (13.6 GHz) radar. The GPM DPR detection performance are slightly improved compared to the TRMM precipitation radar (PR) with Minimum Detectable Signal (MDS) of 14.5 dBZ at Ku and 16.3 dBZ at Ka in the matched scan (MS) mode (Hamada and Takayabu, 2016). The inclusion of a second frequency in GPM has already demonstrated improvement



in many aspects such as the ability to retrieve parameters characterizing the DSD in rain (Gorgucci and Baldini, 2016) and value in improving the rain classification (Le et al., 2016). Moreover, coincident measurements from the CloudSat CPR and the GPM

DPR of the same precipitating system have illustrated that cm and mm-radars are effective in mapping different parts of the precipitating system and can be used synergistically in order to better retrieve cloud microphysical properties. Following the guidelines provided by the 2017 NASA Decadal Survey, multi-frequency Doppler radars, with different combinations of Ku, Ka and W bands, have been proposed as the core instruments of the  Aerosol Cloud Convection and Precipitation (A-CCP) mission GHz) (Kummerow et al, 2020, Battaglia et al., 2020b).  Multi-frequency radar observations are especially valuable in ice/snow

cloud conditions because of the large variability in scatterers' microphysical properties (e.g. particle size, shape and density). The use of multi radar frequencies of which at least one is in or close to the Rayleigh regime (cm wavelength) and one is sufficiently affected by non-Rayleigh scattering (mm wavelength) has been proposed to  improve retrievals of cloud properties over single-frequency applications (section 2). Better understanding of ice cloud characteristics and composition will relax assumptions made on the retrieval of precipitation rate of ice (von Lerber et al., 2017), and ice water content (IWC) which is

needed to understand the global distribution of the ice-phase precipitation and, therefore, enhancing our knowledge of the global water and energy budget.

Despite the valuable information the existing space-borne systems have been providing so far, gaps in the detection and characterization of precipitation remain, especially when the capabilities in multi-frequency radar observations of ice/snow are considered (Battaglia et al, 2020b). To date, very few airborne experiments (e.g. the 2003 Wakasa Bay Advanced Microwave

Scanning Radiometer Precipitation Validation Campaign (Lobl et al. 2007), and the 2015 Olympic Mountains Experiment (OLYMPEX) (Houze et al., 2017)) collected triple-frequency radar observations but only with limited coincident airborne in situ cloud microphysical data (e.g. the OLYMPEX provides 2.2 hours of in-cloud data with Ku-Ka-W radar data and coincident microphysics, Chase et al., 2018, Tridon et al., 2019). To the best of our knowledge, there are no publicly available such coincident datasets from high-latitude regions where precipitation is dominated by shallow, low intensity, snow or mixed-phased

precipitation.

The RadSnowExp (Wolde et al., 2019) is a multi-platform and multi-sensor study organized by the European Space Agency (ESA) and conducted by the National Research Council of Canada (NRC) and Environment and Climate Change Canada (ECCC) to address the pressing need for provision of precipitation measurements, locally and globally. The research flights were conducted in mid-latitudes and near the Arctic circle (Iqaluit, NU, Canada, ~63N), during the fall of 2018, covering a wide

geographical region and microphysical conditions, at a temperature range -50 to 5 °C and altitude extending to 7 km (Fig. 1). The flights focused on sampling precipitation systems where large aggregates and rimed particles were present in order to optimize the triple-frequency analyses. Multi-frequency radar observations were carried out by the NRC Airborne W and X-band (NAWX) radars and the University of Wyoming's Ka-band Precipitation Radar (KPR) (Haimov et al., 2018). In addition to the radars, the NRC Convair-580 aircraft was equipped with extensive in-situ and remote sensing sensors installed in various

locations of the aircraft, including on the underwing and wingtip pylons, various locations of the fuselage, and inside the aircraft cabin (Fig. 2). The dataset collected in flight during the RadSnowExp campaign uniquely features:

- collocated, high resolution triple-frequency radars with near coincident in situ measurements;
- state-of-the-art in situ sensors covering the whole scale of atmospherically relevant hydrometeor diameters, from aerosol size to precipitation size, along with high resolution imaging probes for single-particle identification.

- complementary measurements of atmospherics state parameters and cloud phase detection.



In this study, airborne measurements are used to evaluate findings from recent multi-frequency radar modelling studies on the multi-frequency radar signatures of ice particles of varying habits, shapes, and sizes in different precipitation systems including intensive snow events in the mid and high-latitude regions.

This paper is structured as follows. Section 2 details on theoretical studies of triple frequency / multi-frequency. In section 3, airborne data processing and methodology for the airborne triple-frequency analysis are described. In section 4, the experimental evaluation of triple-frequency study using the RadSnowExp dataset is presented. Finally, conclusions and discussions are given in section 5.

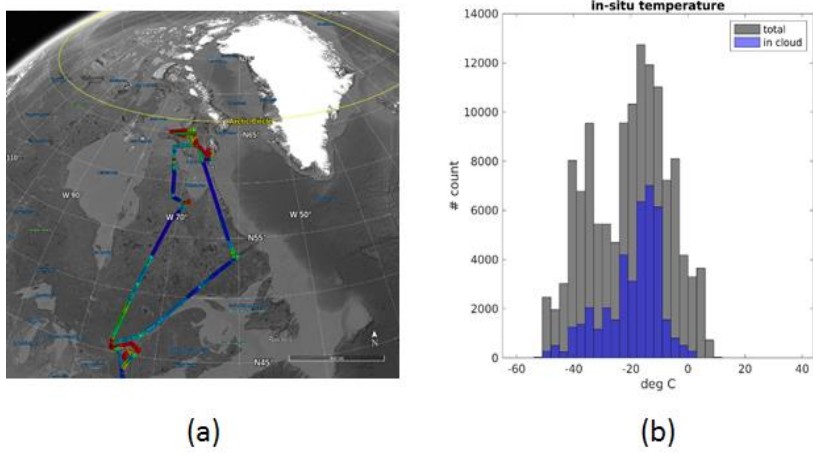

**Figure 1: Full flight path of the RadSnowExp campaign (a) and a histogram of the temperature range and frequency encountered in RadSnowExp flights (b).**

## 2 Multi-frequency radar ice retrieval potential

Multi-frequency radar observations are especially valuable in ice/snow cloud conditions. Ice crystals complexity and large variability in microphysical properties (e.g., density, size, shape) make the interpretation of single-frequency radar observations extremely challenging. The rationale for multi-frequency radar observations is detailed in recent papers (Ori et al., 2020; Battaglia et al., 2020a-b). In this section, we summarize some of the key results.

When comparing measurements of reflectivities from two radars operating at different frequencies $f_1$ and $f_2$ ($f_1 < f_2$) it is possible to consider the dual frequency ratios (DFR), defined as their difference in logarithmic units (equivalent to their ratio in linear units),

$$DFR_{f_1/f_2}(r)(dB) = Z_{f_1}^m(r) - Z_{f_2}^m(r) = \overbrace{Z_{f_1}^{nr}(r) - Z_{f_2}^{nr}(r)}^{non-Rayleigh\ effect} + \overbrace{2\int_0^r \left(k_{f_1}(r) - k_{f_2}(r)\right)dr}^{attenuation\ effect} \quad (1)$$

where $Z_{f_1}^m(r)$ and $Z_{f_2}^m(r)$ are measured radar reflectivity factors in dBZ at range $r$ and frequencies $f_1$ and $f_2$, respectively; $Z_{f_1}^{nr}(r)$ and $Z_{f_2}^{nr}(r)$ are reflectivity factors due to non-Rayleigh effect. $k_{f_1}(r)$ and $k_{f_1}(r)$ denote specific attenuation ($dB/km$) at range $r$.

In equation (1), we have highlighted the two possible contributions to the DFR:

- "non-Rayleigh effects", i.e. differences in the effective reflectivity factors of the targets which occur when the hydrometeor sizes are comparable to the radar wavelength (Bohren & Huffman, 1983; Lhermitte, 1990);

- "attenuation effects", i.e. differences in the attenuation properties along the propagation path, with higher attenuations produced at higher frequencies (Lhermitte, 1990, Tridon et al., 2020).

Non-Rayleigh effects result from intensive properties of the PSD (e.g. characteristic size, spread of PSD) whereas attenuation effects can be used to infer extensive quantities (e.g. concentrations, rain rates, equivalent water contents). Because of the variety of ice habits and shapes, the computation of scattering properties of ice crystals is much more complex than for raindrops (Kneifel et al. (2020) and references therein); whilst at small sizes backscattering cross sections are proportional to the square of the mass of the crystals (Hogan et al., 2006), when approaching large sizes the mass distribution within the particle along the direction of the impinging radiation plays a key role in affecting the particles scattering properties (e.g. Hogan & Westbrook (2014)). An example of DFR calculations for exponentially and Gamma-distributed ice crystals is shown in Fig. 2 where points diverge from the origin, which corresponds to the Rayleigh approximation when moving to larger sizes. There is clearly a large variability in the triple frequency observables introduced by the different shapes and degree of riming of the ice crystals as thoroughly demonstrated in Mason et al., 2019.

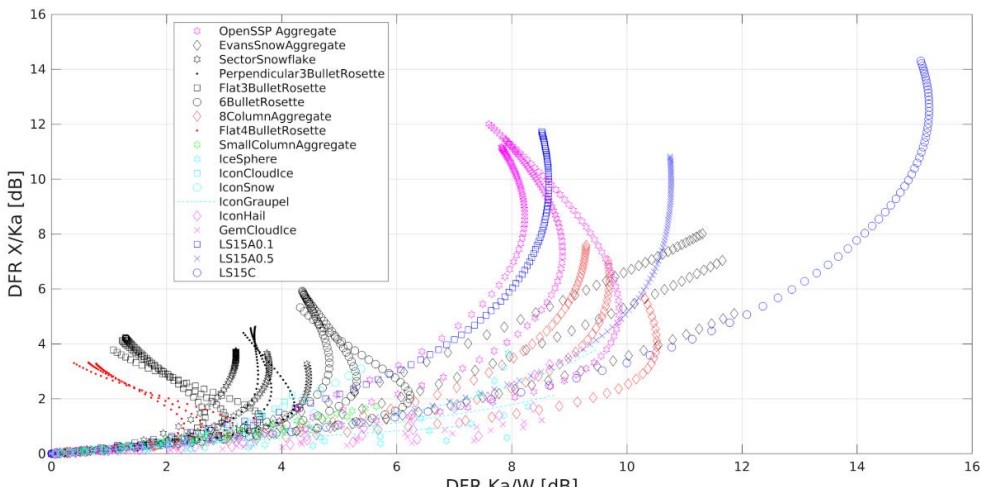

**Figure 2:** Example of DFR Ka/W vs DFR X/Ka corresponding to different populations of snow habits with different characteristic diameters of PSD. The habits correspond to state of the art scattering models: the first habit is a mixture of aggregates from the database described in Kuo et al., 2016; the next 14 habits are extracted from the ARTS scattering database (Eriksson et al., 2018); the last three habits are from the models of Leinonen and Szyrmer, 2015. For the first two classes of models scattering properties are computed via discrete dipole approximation for Gamma-PSD with µ equal -2, 0 and 8 (same symbols); for the last class the self-similar Rayleigh Gans approximation (for the corresponding coefficients see details in Mroz et al., 2021a) is used with exponential PSDs.

## 3 Data and methodology

### 3.1 Airborne radars

In this study, triple frequency radar data from the NRC airborne W- and X-band radar (NAWX) and the Wyoming K-band Precipitation Radar (KPR) measurements from nadir and zenith looking antennas are used. The NAWX antennas are housed inside an unpressurized blister radome mounted on the right side of the aircraft fuselage (Fig.3a) and the KPR radar was installed on the left wingtip pylon. Some important radar parameters are given in Table 1. More detailed information on the NAWX radar system and KPR can be found in Wolde and Pazmany (2005) and Haimov et al. (2018), respectively. In the RadSnowExp



project, the radar complex I and Q samples are processed to powers and complex pulse pair products according to the radar parameter specifications table. These products are recorded in binary format. Although the three radars are almost collocated,

additional signal processing steps are needed to provide highest level of radar volume matching to reduce the DFR estimation errors and to provide the best evaluation of the radar measurements in synergy with in situ microphysics observations.

Table 1: Radar parameters for the RadSnowExp campaign.

| Parameter | W-band | Ka-band | X-band |
|---|---|---|---|
| RF output frequency | 94.05 GHz | 35.64 GHz | 9.41 GHz ± 30 MHz |
| Nadir/Zenith antenna beamwidth | 0.75 | 4.2 | 4.5° |
| Pulse width | 500 ns | 250 ns/2.5 µs or 500 ns/5 µs (short pulse/chirp) | 500 ns |
| Range resolution | 75 m | 30 m 60 m | 75 m |
| Dwell time | 0.14 s | 0.2 s | 0.23 s |
| Sampling resolution | 17.13 m or 34.26 m | 15 m or 30 m | 30 m |

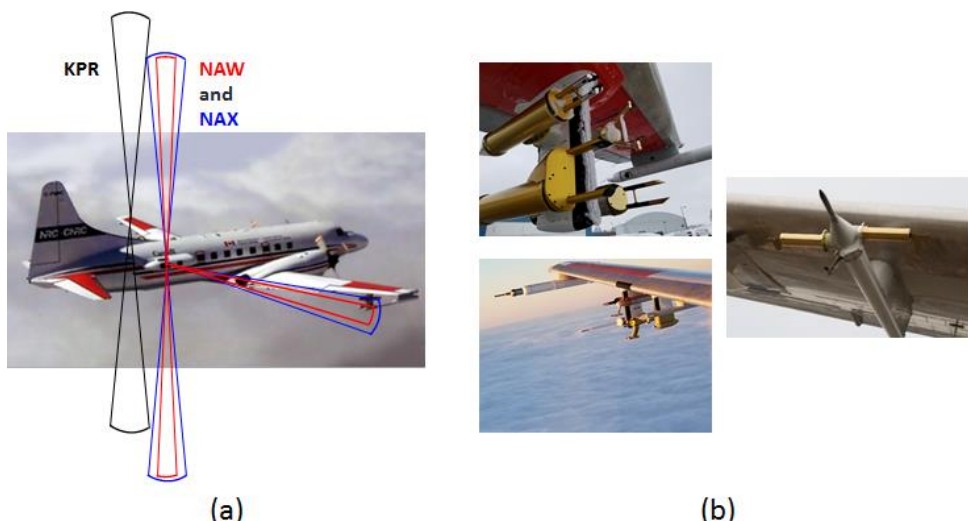


**Figure 3: Locations and direction of NAWX and KPR radars and antennas beams (a) and wing-mounted microphysics sensors and air data probes (b).**

### 3.1.1 Radar data volume matching

To obtain accurate estimates of DFR, radar reflectivity observations at each frequency would optimally sample the exact same

volume; that is, the observations would have perfectly matched horizontal and vertical resolutions, and would be obtained simultaneously. This is not the case with the RadSnowExp dataset due to mismatched radar beamwidths, vertical resolutions and radar data dwell times. Hence, additional processing steps are needed to mitigate these mismatches. The 3 dB beamwidths and





vertical sampling of the Ka- and X- band radar are 4.2º / 30 m and 4.5º / 30 m, respectively, whereas those of the W-band are 0.75º / 34.26 m. The volume matching procedure is described in the following steps.

● Re-alignment of data along the range axis: during aircraft rolls, distance from KPR (mounted on the aircraft wingtip) to the radar volume can be slightly different from that of NAWX. Re-aligning radar data along the range axis is needed. The re-alignment of the KPR data with the NAWX radar was done by using the ground as a reference point. We observe in most cases range alignment for KPR is within 30 m.

● Smoothing: this step is done to reduce the effect of the beamwidth and vertical sampling mismatch. At close range (< 500 m) where radar resolution volumes are small, we assume that the condition of uniform beam filling is met. First, a boxcar average filter with window length of 6 radar samples is applied to NAW data along the time axis. Resultant NAW data will have an effective beamwidth of 4.5º along the flight path which is close to that of NAX and KPR radars. Secondly, three radar data are mapped into a common range axis with a resolution of 35 m which is close to the vertical sampling of NAW. Next, measurements from the three radars are temporally averaged to 0.5 seconds. Vertical profiles were recorded every 0.14, 0.23, and 155 0.2 seconds for NAW, NAX and KPR, respectively, and then averaged in post-processing to one profile every 0.5 seconds. This simple smoothing algorithm would mitigate the volume mismatch due to the radar location differences (the NAW and NAX radars are collocated but the KPR is about 10 m away) given an assumption of reasonable homogeneity of the scatterers within a few hundred meters around the aircraft. Finally, collocated triple frequency radar data are binned into a common grid of 0.5 seconds x 35 m (time - range) or 50 m x 35 m at the Convair average ground speed of 100 m s$^{-1}$.

**3.1.2 Radar close range calibration fine tuning**

Calibration for NAWX nadir antennas is made using clear air observations of the water surface backscatter cross section (Li et al., 2005). Calibration for other NAWX antennas and KPR is done by comparing measurements between antenna ports. More details on calibration and results for NAWX and KPR radars are described in Wolde et al. (2019) and Nguyen and Wolde (2021). Figure 4 shows examples of radar vertical reflectivity profiles from nadir antennas for a RadSnowExp flight on 22 November 165 2018. At those sampling times, data from in situ imaging probes (not shown) indicate that the aircraft sampled a region of small ice particles with median volume diameters (MVD) less than 300 μm which is in the Rayleigh scattering region of the three radars (see Table A1 in Battaglia et al., 2020b), i.e. all equivalent reflectivity factors from the three frequencies should be the same. However, it can be seen that, at distances within 700 m from the aircraft, there is disagreement between the measurements and the mismatches become larger at closer ranges. This is explained by the limitations of the radar hardware that affects the 170 measurements at this range, within a few first pulse lengths. For this study, it is critical to obtain reliable radar data that are as close as possible to the aircraft so that the radar and the in situ sensors sample nearly the same volume. In addition, at near distances, the effect of radar attenuation on the radar reflectivity is minimal. Within a couple of hundred meters, radar attenuation at Ka and X band in snows/ice clouds is negligible. W-band attenuation caused by atmospheric gases, water vapour and ice scattering in snows/ice clouds would be also minimal at a distance < 300 m. Data at the first few range gates in the far field 175 distance of the radars, collected in regions of small ice particles, were used to compare the W band to the X and Ka band. Results show that, at a range of 245 m, 1) W-band data is usable (not affected by close range biases); 2) the relative offsets between W-X and W-Ka are nearly constant for each flight. When those offsets are removed, dual frequency ratio (DFR) estimates should be unbiased. The DFR uncertainty in calibration is estimated to be less than 0.5 and 0.8 dB for DFR Ka/W and DFR X/Ka, respectively.

Figure 10d shows reflectivity observations at 245 m away from the aircraft after correcting for close range biases for the entire flights on 22 November 2018. It was verified that 1) relative radar calibration is good across the whole flight (excellent





agreements of triple frequency reflectivity in regions of small particles); 2) radar volume matching and time synchronization are good (fine scale features were consistently captured by three radars). It is noted that at 245 m distance below -5 dBZ, KPR signal becomes noisy, due to the system's low sensitivity so are excluded in the analysis.


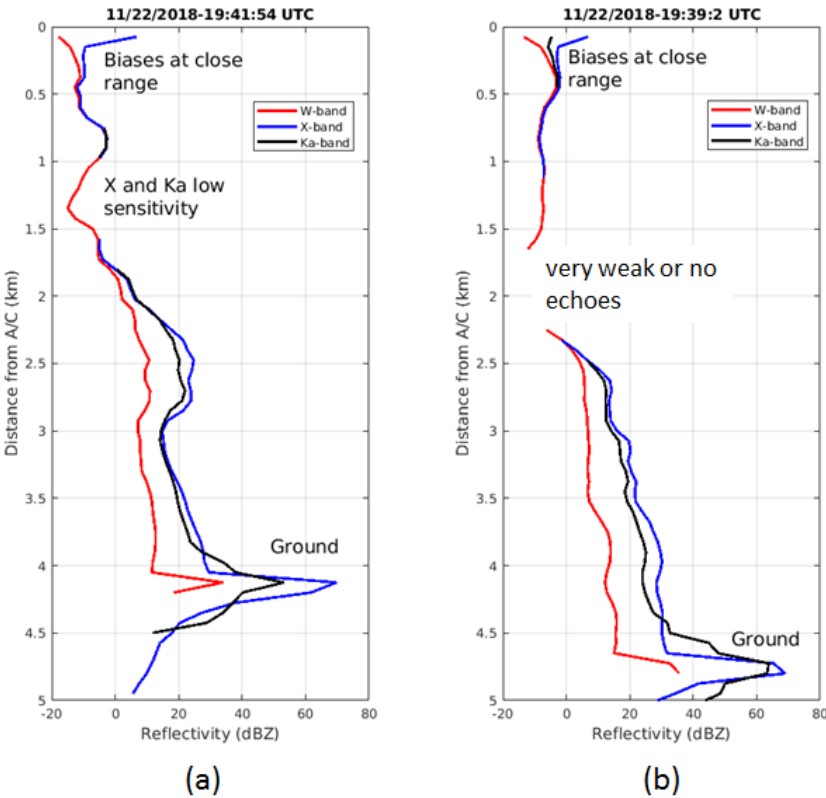

**Figure 4: Examples of vertical profiles from three radars showing different scattering regimes and the mismatch of triple-frequency measurements at close ranges.**

### 3.2 In situ sensors

For the RadSnowExp project, the NRC Convair-580 aircraft owned and operated by the NRC, was jointly instrumented by NRC and Environment and Climate Change Canada (ECCC) with state-of-the-art in-situ sensors for measurements of aircraft and atmospheric state parameters, and cloud microphysical properties. Bulk liquid water content (LWC) and total water content (TWC) were measured simultaneously with particle images and size distribution, ranging from small cloud droplets to large precipitation hydrometeors.

For this work, cloud particle size distribution was composed using a combination of data from several single-particle probes: Fast Cloud Droplet Probe (FCDP, 2-50 μm, SPEC Inc.); two-dimensional stereo (2DS, 10-1200 μm, SPEC Inc.) probe; High Volume Precipitation Spectrometer version 3 (HVPS3, 150-19200 μm, SPEC Inc.) probe or Precipitation Imaging Probe (PIP, 100-6400 μm, DMT) . In addition, Cloud Particle Imager (CPI, 10-2000 μm, SPEC Inc.) provided high resolution (2.3 μm) grayscale imagery of small cloud and drizzle drops, ice particles and portions of large drops and ice crystals and broken large ice

particles. The high resolution of the CPI probe allows identification of riming levels on ice crystals. In order to aid the



determinations of triple frequency radar signatures of various particle compositions and level of riming, the CPI images were classified into 24 different hydrometeor types using machine learning with Convolutional Neural Network method (similar to Praz et. al., 2018) based on a training dataset created from recent projects conducted using the NRC Convair-580. For this paper, we combined some of the classifications and reduced the grouping to nine different types (Table 2). CPI data integrated over 5 seconds are used to compute and to plot the fractions of sampled particle types. For each study case, we present two CPI particle fraction plots, one for all 9 groups listed in Table 2 and one for a subset of ice habits only.

Table 2: CPI classification grouping definitions

| Merged group | Ice particle types |
|---|---|
| Pristine | Columns, capped columns, bullets, bullet rosettes, plates, |
| Dendrites | Stellar dendrites, blurred dendrites |
| Rimed dendrites | Rimed dendrites |
| Rimed particles | Graupels, densely rimed, rimed columns |
| Aggregates | Aggregate columns, aggregate planars |
| Other ice particles | Two-drops, blurred ice, broken triangle, ice, melting large, semi-spheroid, tiny ice |
| Small particles | Particles $< 40 \, \mu m$ |
| Drops | Drops, blurred drops |
| Artifact | Artifact |

Particle size detection range of each probe is illustrated in Fig. 5. The probes were calibrated with glass beads and a spinning chopper before the campaign and re-evaluated in NRC's altitude icing wind tunnel after the campaign. The uncertainties in sizing and concentrations were less than 5%. Taking into account image corrections and rejections, the propagated uncertainties can grow within the range presented by Baumgardner et al. (2017). The single-particle data are then used to derive size distributions and bulk cloud properties.

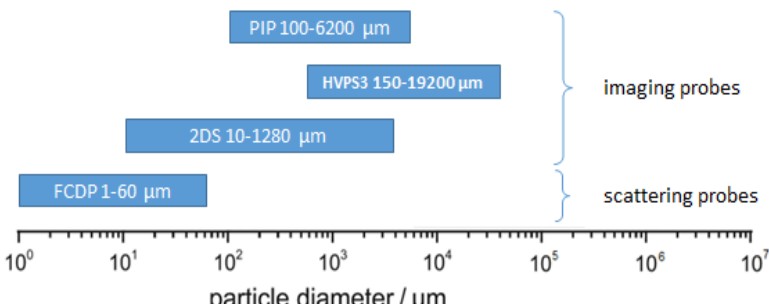

**Figure 5: Single particle detection ranges of FCDP, 2DS, HVPS3 and PIP, jointly covering the broadest detectable range of single particle diameters.**

TWC and LWC were measured by the Nevzorov, a constant-temperature, hot-wire probe (Korolev et. al, 1998). The sensitivity of Nevzorov is estimated to be up to $0.002 \, g \, m^{-3}$ (Abel et al., 2014). Additionally, the Nevzorov ice water content measurements can be subject to increased uncertainty when large hydrometeors are present. It should be noted that the probe is analog and thus prone to artifact originating from aircraft wiring. However, this noise was simultaneous in collector and





reference wires and thus had little effect on the estimated water content. We estimate the accuracy of the Nevzorov data during RadSnowExp to be on the order of $0.05 \; g \; m^{-3}$.

Additionally, the composite PSD, derived from single particle probes, is used to calculate characteristic sizes (Median Volume Diameter - MVD), and concentrations ($N_t$). We set a 50 µm lower bound for N(D) in calculating total volume within PSD, and MVD. This will minimize the impact of supercooled drops in the calculations and interpretation of parameters characterizing ice particles. The exclusion of small particles does not have a major impact on the calculated bulk microphysics (i.e. bulk density and MVD) and radar reflectivity, which are dominated by large particles. Several bulk microphysical parameters calculated from

measured PSD are given below.

- *Effective bulk density* ($\rho_e$) is the ratio of the mass of ice to the total volume of ice within a sample volume. An empirical method to compute $\rho_e$ from PSD (Heymsfield et al., Chase et al., 2018) is defined as,

$$\rho_e = \frac{m_{IWC}}{V} \tag{2}$$

where $m_{IWC}$ is the mass of ice inferred from power dissipated on TWC and LWC sensors of the Nevzorov probe (Korolev et

al., 1998) and $V$ is calculated as the sum of the volume of all particles within the PSD. Here, each particle is approximated as an oblate spheroid with an aspect ratio of 0.6 (Hogan et al., 2012). Both $m_{IWC}$ and V are computed for $1 \; m^3$.

- *Median volume diameter (MVD)* is defined as the diameter for which the total volume of all drops having greater diameters is just equal to the total volume of all drops having smaller diameters. This is the characteristic diameter that contributes most to cloud liquid water or mass. Calculation of MVD is described in Leroy et al. (2016).

- *Particle number concentration ($N_t$)*:

$$N_t = \int_{D_{min}}^{D_{max}} N(D) \, dD \tag{3}$$

In order to show how often we encountered regions that are interesting targets for triple-frequency radars, the complementary cumulative distribution functions of MVD from PSD are calculated for three of the RadSnowExp project flights (22, 25 and 28 November 2018;Fig. 6). Among the three flights, two of flights were carried out in the Arctic and the one of the flight was

conducted in mid-latitude. It can be seen that in one of the flights (the 22 November, 2018), 13% of the sampling points were in the non-Rayleigh region for all three frequencies whilst there were almost surely no similar sampling point for the other arctic flight (25 November in Fig. 6). For this reason, the analysis in this paper is focused on the arctic flight conducted on 22-November 2018.

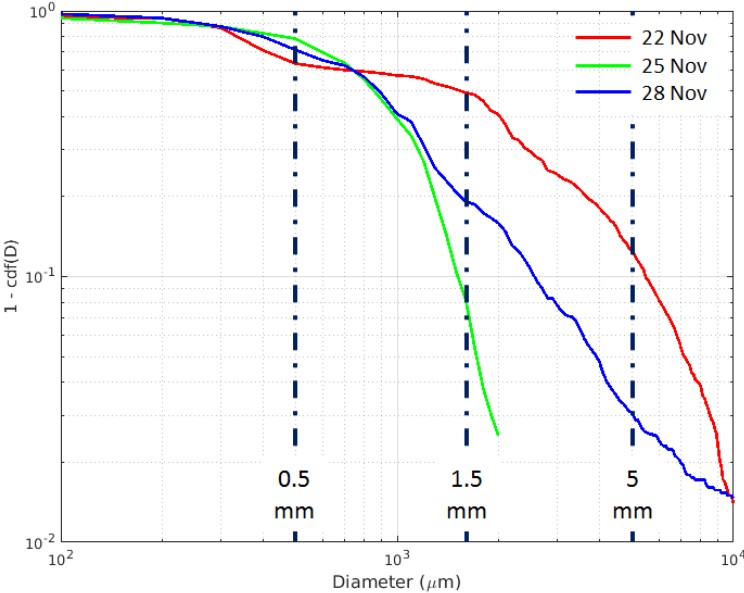


**Figure 6: The complementary cumulative distribution functions of particle diameter from PSD for flights on 22, 25, and 28 November 2018. As a rule of thumb non-Rayleigh effects of the order of 1 dB appear for diameters exceeding 0.5, 1.5 and 5 mm at W, Ka and X, respectively (see Table A1 in Battaglia et al., 2020b).**

**3.3 Collocating radar and in situ measurements**

Collocating radar and in situ measurements is a critical step for accurate determination of relations between microphysics and radar scattering properties. Coincident measurements and perfectly matched volumes would provide the most accurate assessment. However, in reality, radar sampling volumes are much larger than those of cloud probes and both sample volumes are not spatially collocated. In literatures, collocating radar and in situ data is often archived by averaging radar data over a large time such as in ground based observations or alternatively finding nearest airborne radar data points to the in situ. For example,

in Chase et al. (2018), radar and in situ data were obtained from two different platforms and post-processing algorithms assumed that radar volumes within 10 min temporally and 1 km spatially of in situ were considered collocated. Moreover, the in situ observations were assumed to be characteristic of the entire matched radar volume despite the differences in the radar and probes sample volumes.

In our case, the radars and in situ probes are in the same platform and share a common GPS time server so their data are

temporally synchronized. Temporal sampling rate of the post-processed triple-frequency radar data is 0.5 seconds (section 3.1a). For particle probes, data are usually integrated over a period of 2-5 seconds for good quality; hence, the radars need to be decimated to match with the in situ. On the other hand, there is a difference in sampling location between the radar and in situ. The nearest reliable NAWX and KPR radar data for triple-frequency analysis is 245 m above or below locations where in situ data were measured (section 3.1b). Although the setup offers much higher accuracy in radar - in situ measurement coincidence

compared to previous studies, it still brings in a question of how the radar data should be processed along the range axis to best characterize the microphysics. In order to answer that question, first, we need to examine the variability of DFRs in the range dimension. This is done using data from several flight segments during the RadSnowExp campaign.





### 3.3.1 DFR variability

The DFR variability studied in this section is defined as the fluctuation in DFR values along the radar range axis and will be

analysed by comparing DFRs computed above and below the aircraft. Figure 7 shows examples of scatter plots of DFRs at the first usable distance (245 m) above and below the aircraft for all data points in a RadSnowExp flight on 22 November 2018. In the region of DFRs < 5 dB, the difference of DFRs at the two directions is often within 2-3 dB but for DFRs between 10 and 15 dB the difference can be as large as 8-10 dB.

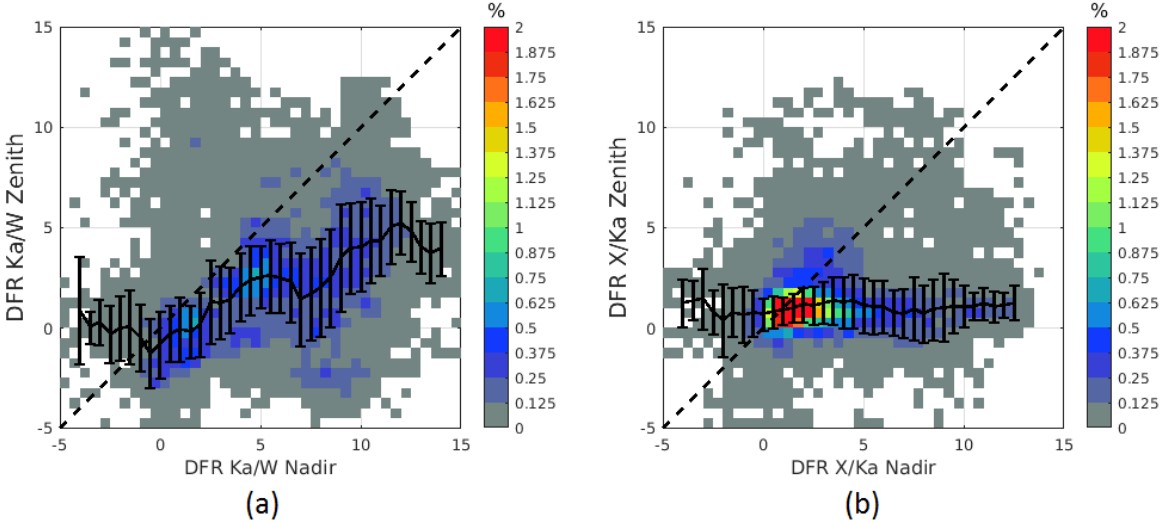

**Figure 7: Scatter plot of DFR Ka/W (a) and X/Ka (b) of radar profiles 245 m above and below the aircraft for all data points from a RadSnowExp flight on 22 November 2018.**

### 3.3.2 Data selection

The DFR variability study in the previous section shows that at a given time, reflectivity ratios between two frequencies could vary up to 8-10 dB within 490 m in altitude, i.e. between the 245 m profiles above and below the aircraft. Averaging radar data

over multiple range gates around the in situ sampling might increase biases in DFR estimates; thus, in this study we just use measured DFRs nearest to the in situ sampling. The remaining question is which dataset, above or below the aircraft, should be selected. In order to assess how well the radar data would match the measured particle size distribution (PSD), equivalent reflectivity factor at X-band is forward modelled from the measured composite PSD using the Rayleigh-Gan spheroidal approximation and Brown and Francis (1995) mass size relation. The X band is chosen because it is least affected by attenuation

and non-Rayleigh scattering effect. The simulated X-band reflectivity is then compared to the NAX radar data using the Pearson's correlation coefficient. Correlation coefficients using a 10 minute long running window are used to determine flight segments to be analysed. Specifically, only data points with correlation coefficients higher than a 0.6 threshold were selected in the analysis. This will ensure an accurate analysis of the triple-frequency radar. Illustrations of this procedure are given in Fig. 10c.





### 4 Triple-frequency case study: Arctic storm on 22 November 2018


On 22 November 2018, the Convair-580 conducted a flight in the Canadian Arctic across the Frobisher bay area near Iqaluit. Spiral and lawnmower patterns were used for sampling at the outskirts of an Arctic storm which is clearly visible on the imagery from AVHRR sensor (channel 4, 10.3 $\mu$m) on board NOAA 13 polar orbiting meteorological satellite (Fig. 8a). At the beginning of the flight, the aircraft climbed to 6 km and later descended to 1.7 km in steps. Next climb was in steps, to 2.9 km, followed by

descent and landing (Fig. 8b). In terms of cloud properties, mostly mixed-phase conditions were observed, with moderate to heavy icing causing electrostatic discharges on the windshield in the second half of the flight. Diverse hydrometeor habits including rosettes, rosette-aggregates, and irregular shapes were sampled during the cruise at an altitude of ~6 km with in cloud temperature of -40 °C (Fig. 9) whereas pristine plates, capped columns, densely rimed particles were observed at the lower altitude with temperature centered around -15 °C. Figure 9 shows the ground to air temperature and the distribution of the in situ

temperature for flight 22 November.

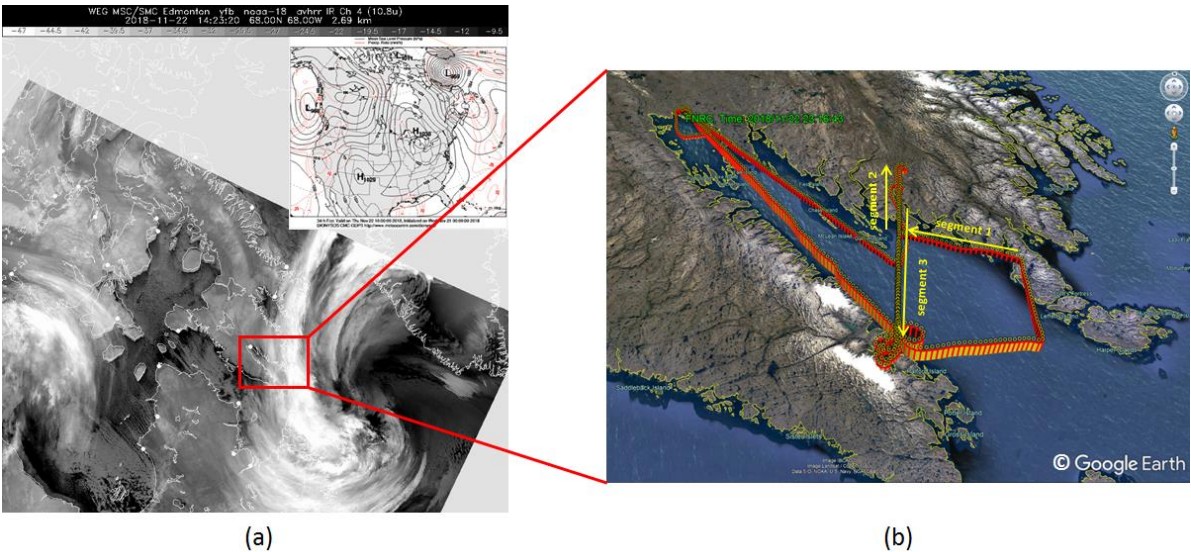

(a)                                                                    (b)

**Figure 8: NOAA-13 10.3 µm channel AVHRR imagery showing the Arctic storm (a), and the Convair flight track on November 22nd (b). Locations of three legs of this flight used for the case studies are indicated in (b).**






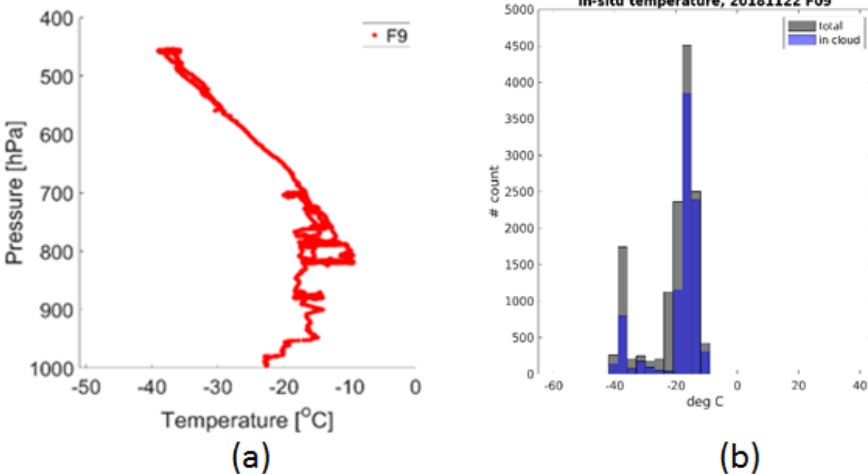

**Figure 9: During the 22 November flight, the ground to air temperature was in a range of [−40℃, −10℃] (a). A temperature inversion was located at 1 km altitude. Histogram of the in situ temperature (b) shows that for most of the flight the in cloud temperature was at around −15℃.**


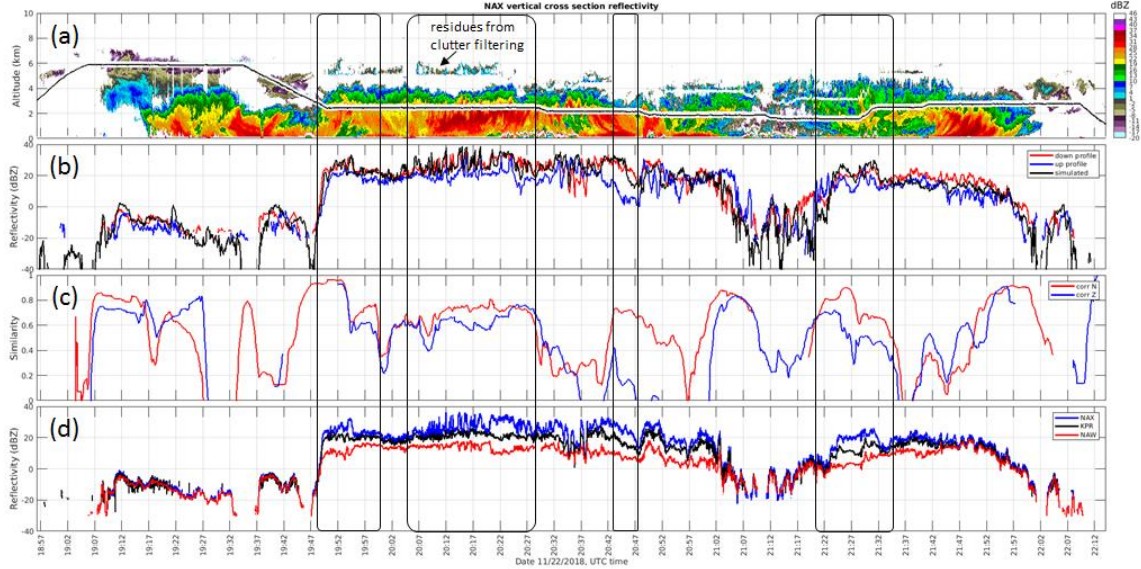

**Figure 10: X-band vertical cross section reflectivity for the flight on 22 November (a); reflectivity profiles at 245 m above and below the aircraft along with simulated X-band reflectivity from measured PSD (b); similarity measurements (correlation coefficients) between above/below Z profile and simulated Z profile (c); reflectivity profiles of X-, Ka-, and W-band radar at 245m below the aircraft showing the regions with interesting triple-frequency radar signatures (d). Boxes indicate the specific segments that will be analysed further.**


In Fig. 10a, vertical cross section reflectivity at X-band of the entire flight is shown. The X-band data is selected to be

representative of the radar reflectivity vertical structure of the storm as it is the least affected by attenuation and non-Rayleigh

scattering. It is noted that there is a gap in the data at the up antenna of the X-band radar and some residues from the filtering of

ground clutter leakages (Nguyen et al., 2021). In addition to the radar time-height reflectivity cross section, radar reflectivity





profiles at a distance of 245 m from the aircraft at up and down antennas are depicted along with simulated X-band reflectivity using the in situ PSD data. The probability density functions (pdf) of the X-band reflectivity at the nearest range above and below the aircraft are shown in Fig. 11. The pdf figures show that the aircraft stayed in inhomogeneous cloud layers (as highlighted by the difference between the nadir and zenith data with higher reflectivities typically occurring below the aircraft).

The correlation coefficients between simulated and measured X-band reflectivities (section 3b), as functions of time, are shown in Fig. 10c. For this flight, data from the down antenna often have higher correlation with the in situ data than data recorded at the up antenna. Radar data with correlation coefficients ≥ 0.6 would be considered to be a good match with the in situ data. In addition to the similarity measurements, reflectivity values and DFRs are also used to select case studies. In this work, we focus on instances where non-Rayleigh scattering occurs as indicated by differences in radar reflectivity measured at the three frequencies (Fig. 10d). We have selected three different segments for further analysis of triple-frequency (indicated by box 1, 2,

and 4 in Fig. 10).

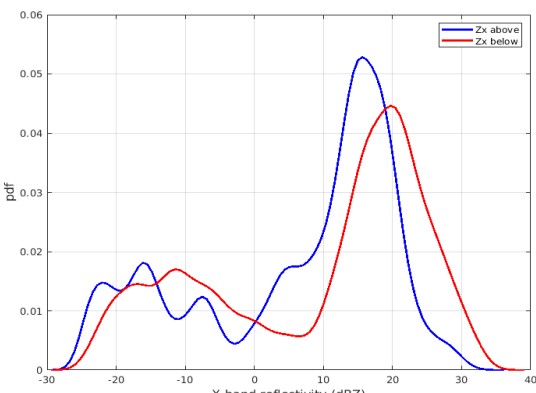

**Figure 11: PDFs of the nearest range reflectivity above and below the aircraft for 22 November 2018 flight.**

**4.1 Segment 1: 1948-2000 UTC**

In this segment, the aircraft descended from 2.8 km to 2.4 km with temperature spanning the range of [-18, -15] °C. During the descent, the aircraft first sampled irregular shape ice crystals and small-ice in a mixed-phase environment with maximum size < 1 mm and then stayed at the same altitude sampling mixed phase clouds consisting of supercooled cloud drops of various sizes, rimed dendrites, pristine ice crystals and irregular types. The case is divided into five different sections (A-E) for detailed analysis based dominant particle compositions that resulted in discernible DFR signatures. Fig. 12 show the average PSD and

mass distribution profiles of the five sections selected for detailed triple-frequency analysis. The PSD and mass distributions are generally bi-modal with two modes around 30 $\mu m$ and 1 mm.



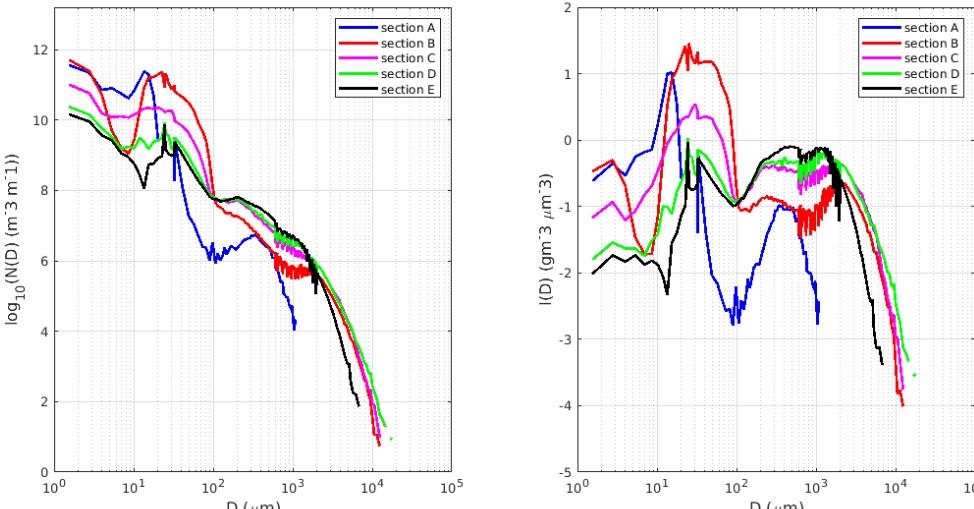

**Figure 12: Averaged PSD (a) and spectral distribution of IWC (b) profiles for five sections selected in Fig. 12. The BF95 mass-size relationship was used to compute the IWC spectrum.**

In Fig. 13, the left panels' show time series of the triple-frequency reflectivity, DFRs, PSD spectrum, MVD, IWC and LWC for this study case. The right panels show the fractional composition of cloud particle types within the CPI detection range (<2 mm) along with representative images of single particles extracted from CPI and HVPS3 for each flight section. In the CPI cloud composition plots, the top panel shows the fractional composition of all major hydrometeor types over time (Table 2), and in the middle panel, only the fractional composition of the ice subset is shown. In the first section (section A), during descent, the most common habits are irregular and small-ice. The DFRs are near 0 dB, in agreement with the small particle sizes shown in PSD and the CPI images. As the aircraft entered into mixed phase clouds at the start of section B at the altitude of 6 km, there was a significant increase in the number of drops and dendrites with some heavily rimed dendritic fragments. Subsequently, bigger aggregates start to appear in the HVPS3 detection range. At around 19:51 UTC, the fraction of drops (Fig. 13, top right panel) increased to its maximum values, which is consistent with the LWC peak (up to 0.2 $gm^{-3}$) observed by the Nevzorov probe (Fig. 13, bottom left panel). With the presence of large particles (dendrites, and rimed particles), the DFR values sharply increased to ~10 dB (Ka/W) and ~4 dB (X/Ka). There are some DFR variabilities in this section due to changes in PSD and particle composition. For example the slight decrease in DFR values around the middle of section B (around 19:51 UTC) remarkably mirrors the decrease in the relative concentrations of dendrites and rimed particles (Fig. 13, middle right panel). Section C is from sampling of the storm when the aircraft descended to 1.7 km and sampled heavily rimed dendrites, large aggregates as observed by the HVPS2 probe and a reduced fraction of number of drops by the CPI probe. It is worth to note that in sections C-E, the percentage of pristine, small particle and drop categories are relatively constant. In this section, the DFRs slightly rise, which is consistent with an increase in dendrites portion within ice habits (middle panel in Fig. 13) with some of them heavily rimed. In section D, mainly heavily rimed, fractured ice and frozen drops are present with bigger aggregates detected by HVPS3. The DFR X/Ka reaches its highest value (~13 dB) exceeding the corresponding DFR Ka/W. Interestingly, this section contains large dendrites with heavy riming and the PSD profile is broader and flatter compared to that of section B-C (Fig. 12). It also shows a slight increase in the larger sizes whilst the fraction of dendrites and rimed particles drops to its lowest level. Lastly, in section E, an increase in the number of smaller particles in pristine shapes like plates, rimed dendrites, frozen drops and also smaller aggregates were detected with the HVPS3. The bulk density is also higher in this section and the MVD





from PSDs are also remarkably stable at about 1.6 mm. The DFR X/Ka and Ka/W are fairly constant around 2 dB and 5 dB, respectively. The reduced DFR values are consistent with a decrease in maximum particle size (Fig. 12a). The small variations in DFR values also agree well with the relatively uniform fraction of cloud particles depicted in the CPI frequency plots.

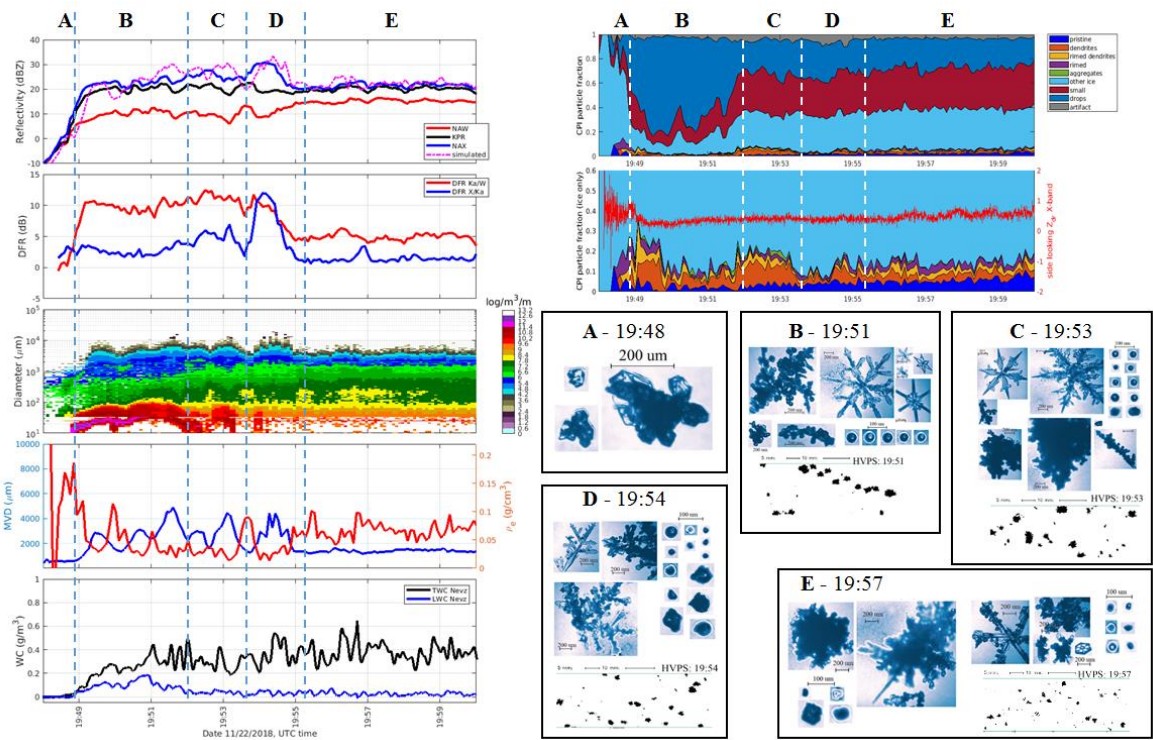

**Figure 13: Left panels from top to bottom: triple-frequency reflectivity profiles; DFR X/Ka and DFR Ka/W; PSD spectrum; characteristic diameters (MVD) and effective bulk density ($\rho_e$); and TWC/LWC from Nevzorov probe for segment 1948-2000 UTC in flight 22 November 2018. Right panels from top to bottom: fractional distribution of all hydrometeors detected with the CPI probes; fractional distribution only of ice habits; representative images from CPI (blue) and lower resolution images of large hydrometeors from HVPS3 (black) for each flight section (A, B, C, D, and E).**

To characterize $\rho_e$, MVD and total concentration ($N_t$) in the DFR plane, the data are presented in Fig. 14 in such a way that each dot represents a data point, the size of the dot being proportional to the MVD with the color corresponding to $\rho_e$ or $N_t$. It can be seen that the DFR values in the five sections (Fig. 13) populate different zones in the DFR plane associated with unique scattering properties of different ice habits. In general, DFRs increase with increasing coincident MVD and DFR X/Ka decreases when bulk density increases. In section B and C where riming occurs, numbers of concentration are significantly higher than

other regions. The section B and C data placement in the triple-frequency plane agrees well with scattering computations of graupel particles using discrete dipole approximation (Fig. 6 in Tyynela and Chandrasekar (2014)). Section D is particularly interesting because of the PSD composition and only aggregate models (Tyynela and Chandrasekar (2014), Kneifel et al., 2015; Stein et al., 2015, Ori et al., 2020) are comparable with the "hook signature" observed in the data points. The distribution of the data points in this section appears as nearly a vertical curve which could be attribute to its broader PSD (Mason et al., 2019). In

this case, we observed large dendritic aggregates with heavily riming clouds. Compared to section C (which overlaps with the scattering computations for spheroid models as in Leinonen et al. (2012)), the total concentration of the data points in section D was much lower whilst the TWC was larger.

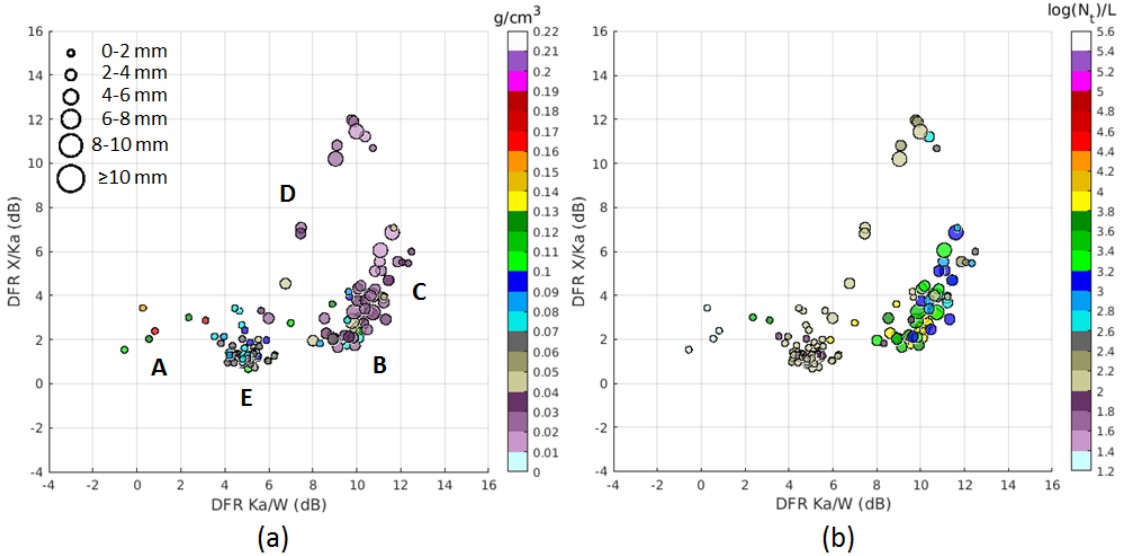

**Figure 14: DFR scatter plots for the 1938-2000 UTC segment, November 22 flight. Data points are coloured by the effective bulk density (ρₑ) (a) and by the total concentration (Nₜ) (b). The dot size is proportional to the calculated MVD. The letters (A-E) indicates each flight section as in Fig. 13.**

### 4.2 Segment 2: 2005-2028 UTC

In this case, the aircraft maintained the attitude of 1.7 km (Fig. 10a), but penetrated regions inland from Frobisher Bay (Fig. 8).

The segment is divided into three sections (A-C) (Fig. 16) for a detailed analysis. This is still a mixed-phase environment with pockets of high concentration of drops (with diameter of approximately 30 μm) and an ice mode at ~1.5 mm (Fig. 15). Different ice habits such as pristine particles, dendrites, fractions of dendritic aggregates, and rimed particles were observed and shown in samples of CPI imagery (bottom right panels, Fig. 16). Larger aggregates (up to 10 mm) were also seen on HVPS3 black shadow images (Fig. 16). It is worth noting that the particle types observed in the three sections are the same. However, the level

of riming and the fraction of dendrites and aggregates within clouds are different between the sections. In section B, the fraction of rimed particles and dendrites (both pristine and rimed) is the highest. The highest TWC ($\sim 1\ g\ m^{-3}$) of the flight was also recorded in section B. During this section, the X-band radar reflectivity increased in land with a number of high reflectivity cores at flight level (Fig. 10, second panel), which is consistent with the high TWC and higher relative concentrations of rimed and dendrites in the CPI frequency plot. In section C, pockets of high TWC were also observed and the fraction of dendrites and

rimed particles remains high with an increase of the relative portion of pristine dendrites and aggregates (Fig. 16, middle panel). The fluctuation in the time series of the observed DFRs matches very well with that of the cloud particle mean diameters (Fig. 16, left panels). It is also consistent with the fraction of rimed particles, dendrites, and aggregates (shown in the CPI composition plot, Fig. 16). In section A, mean values of DFR X/Ka and DFR Ka/W are ~2 dB and ~6 dB with $2\ mm < MVD < 4\ mm$, respectively and DFR X/Ka at times reached the same level as DFR Ka/W at around 8 dB for larger $MVD$ (~6 mm). Also, in

section A, side-looking $Z_{dr}$ fluctuates and, at points, reaches up to 1.5 dB which we suspect to be a result of dendritic particles/needle aggregates dominating the radar measurements. In section B, the DFRs show high variability, mimicking the $MVD$ changes and peaked at ~10 dB for both DFR X/Ka and DFR Ka/W when the TWC is greater than 0.6 g m⁻³ and $MVD$ is greater than 8 mm. The decrease in $Z_{dr}$ in this section is consistent with higher degree of riming (Li et al., 2018). In region C, the





DFR values remain high with the DFR X/Ka reaching over 12 dB. In section B and C, with the increasing number of large

spheroidal compact aggregates due to riming, $Z_{dr}$ is stable at ~0.5 dB.

Distribution of all the data points in this segment in the DFR plane is shown in Fig. 17. Due to a large fluctuation in the DFRs, there are overlapping data points between the sections. Section A is characterized by the presence of small particles, hence it is mainly populated by relatively smaller dots with higher effective bulk density at 1.2 $g/cm^3$. Data points in section B, where the fraction of riming particles reaches its highest value, overlap with both section A and C. Data points in section B and C overlap

but in section B, where the fraction of riming particles reaches its highest value (Fig. 16, left panel), the PSD is flatter (Fig. 15b) and the concentration of small drops is lower (Fig. 15a). In this segment, the location of all the data points shows a very clear illustration of the "hook signature", i.e. the DFR Ka/W values decrease whilst the DFR X/Ka continually increases. Modelling results show a hook signature in triple frequency space from dendritic and needle aggregates (Petty and Huang, 2010), and snow aggregates composed of a variety of different primary crystal habits (Tyynela and Chandrasekar (2014), Leinonen and Moisseev

(2015)) which agree with our in situ observations.

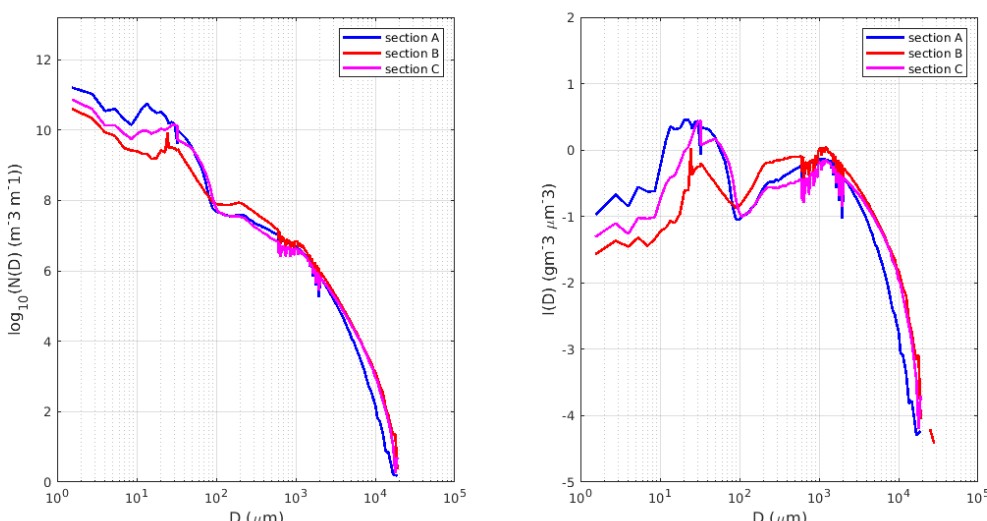

Figure 15: Averaged PSD (a) and mass distribution (b) profiles for three sections (A-C) selected in Fig. 15.





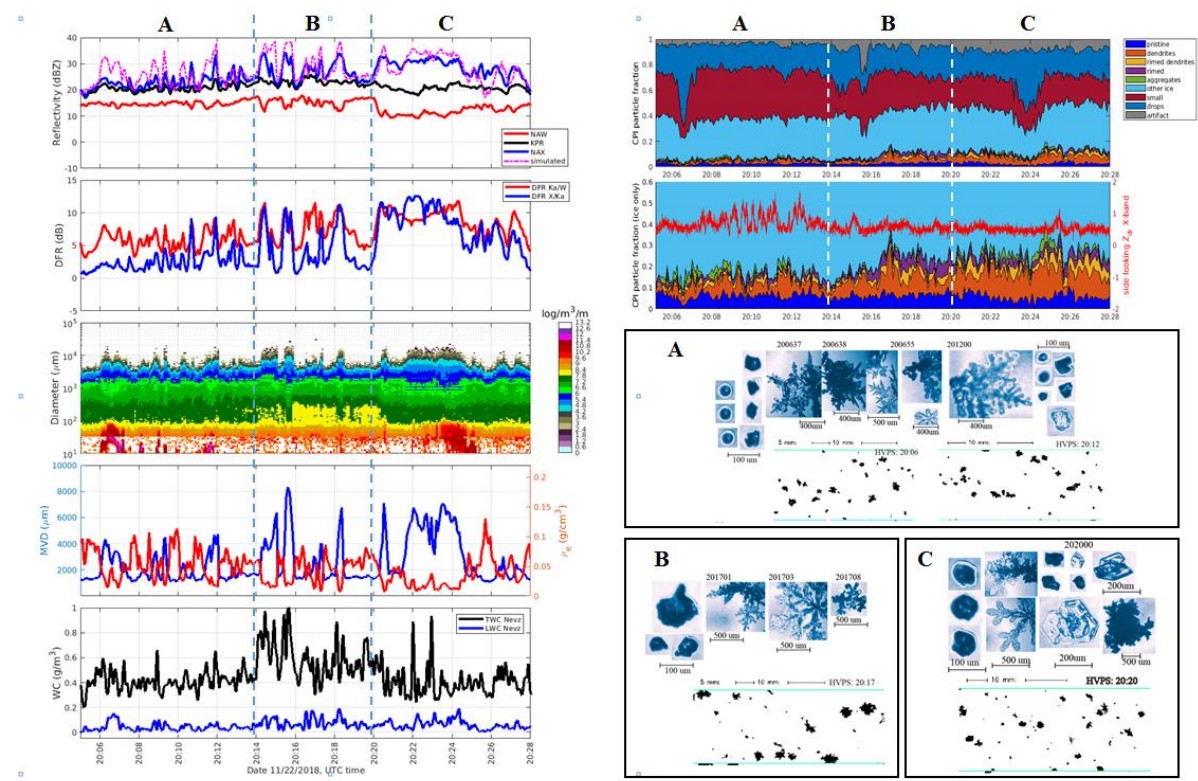


**Figure 16: Similar to Fig. 12 but for flight segment 20:05 UTC – 20:28 UTC.**

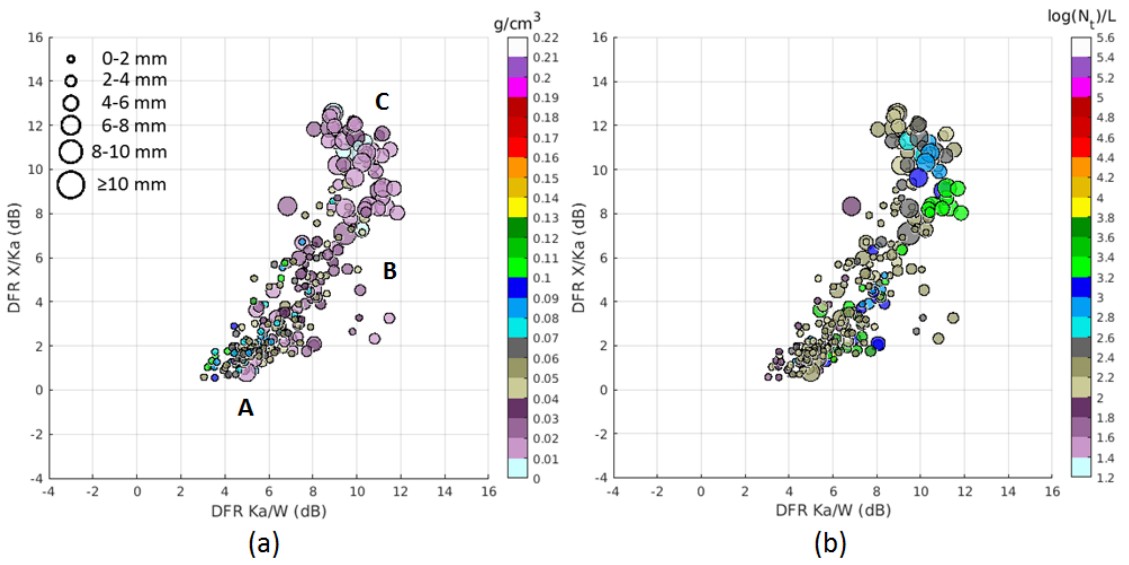

**Figure 17: Similar to Fig. 14 but for flight segment 20:05 UTC – 20:28 UTC.**





**4.3 Segment 3: 2121-2135 UTC**

For this case the aircraft sampled the precipitation system at a lower altitude of 1.7 km (Fig. 10) and then climbed up to 2.2 km at the end of the segment. During this segment there was heavy ice accretion on the aircraft with subsequent electrostatic discharges on the windshield. The segment is divided into four subsections (A-D) based on the DFRs and cloud property signatures (Fig. 19). Section A consisted mainly of supercooled liquid droplets with LWC of ~ 0.2 $g/m^3$ with a small fraction of sector plates and heavily rimed (<2 mm) particles (middle left panels of Fig. 19). DFR X/Ka and Ka/W are, in general, around 2 dB which is

consistent with this type of cloud particles. Effective bulk density and concentration are also at their highest values, ~0.8 − 0.9 $g/cm^3$ and ~$10^{4.5}$, respectively. In section B, supercooled liquid droplets and small ice still dominated but started decreasing while milimetric rimed aggregates with the main mode at ~3 mm (Fig. 18b) appear. Both DFRs increase when the MVD increases with DFR Ka/W filling in the entire range from 2-10 dB and DFR X/Ka reaching up to 5 dB. In section C, more large aggregates with maximum size exceeding 10mm are found. At the beginning of this section (until 21:27 UTC), DFR Ka/W

mirrors the change in MVD. DFR Ka/W goes up to 10-12 dB at MVD ~ 6000um which is similar to the case of section C in the segment 2. However, DFR X/Ka is much lower at 5-7 dB. Moreover, after reaching its highest values (~ 12 dB), DFR Ka/W starts decreasing whilst DFR X/Ka continually increases and DFR X/Ka exceeds DRF Ka/W at ~ 21:27 UTC. Visual analysis of the CPI images reveals the presence of aggregates of rimed dendrites with lower density (Fig. 19) during this period. Also, the PSD in this section is also boarder and flatter (Fig. 19) which affect the distribution in the triple-frequency plane. After 21:27

UTC, DFR Ka/W decreases whilst DFR X/Ka continually increases thus creating a clear hook signature (Fig. 20). The DFR values and patterns agree well with modelling results for aggregate of fernlike dendrites (Tyynela and Chandrasekar, 2014). In fact, the fraction of dendrites in this period is noticeably higher than that at the beginning of the section (top right panel, Fig. 19). In the last section (D), where the aircraft ascended from 1.7 km to 2.9 km, the fraction of rimed particles and aggregates with MVD ~ 1mm increased. The bulk density in section D is higher compared to other sections (left panel, Fig. 19) consistent with

heavily rimed clouds identified from the CPI probe. Both DFRs start decreasing which mirrors a decrease in MVD and become comparable at around 3-4 dB. It is also worth noticing that the concentration in this cloud segment is much higher than in the previous two cases.

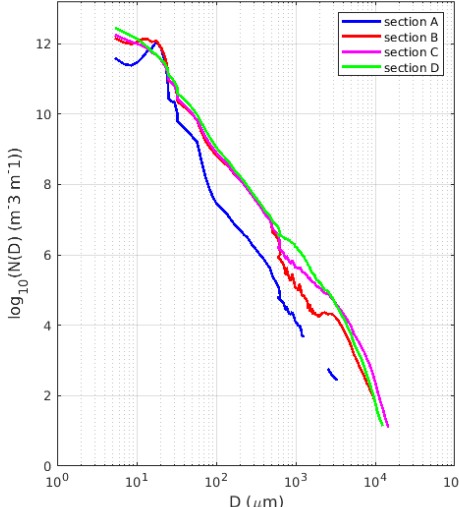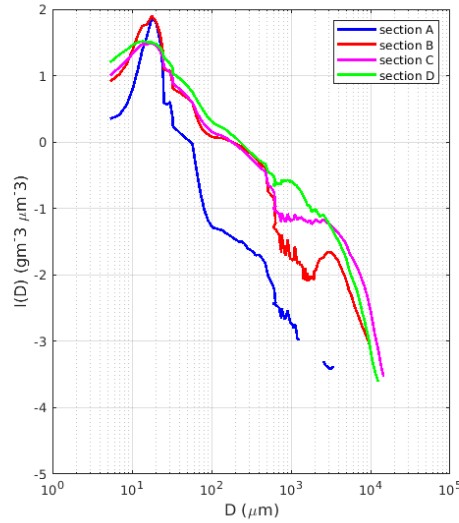



**Figure 18: Averaged PSD (a) and mass distribution (b) profiles for four sections (A-D) selected in the Fig. 19.**

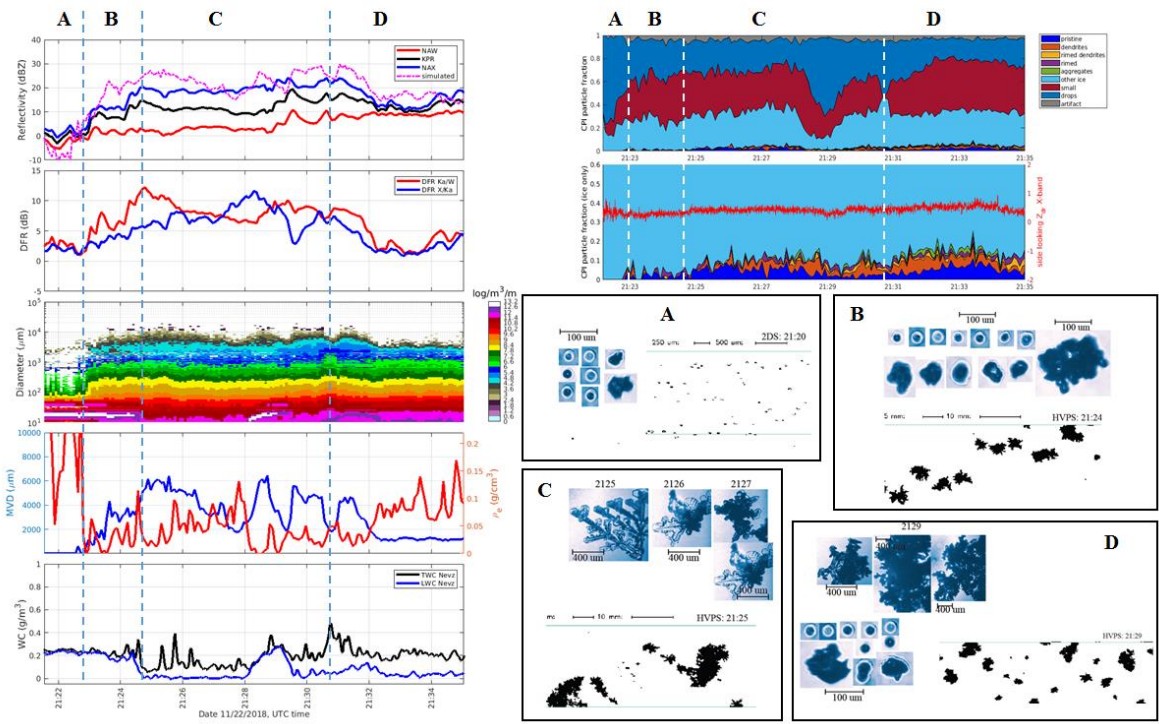

**Figure 19: Similar to Fig. 12 but for segment 21:21:30 UTC – 21:35 UTC.**

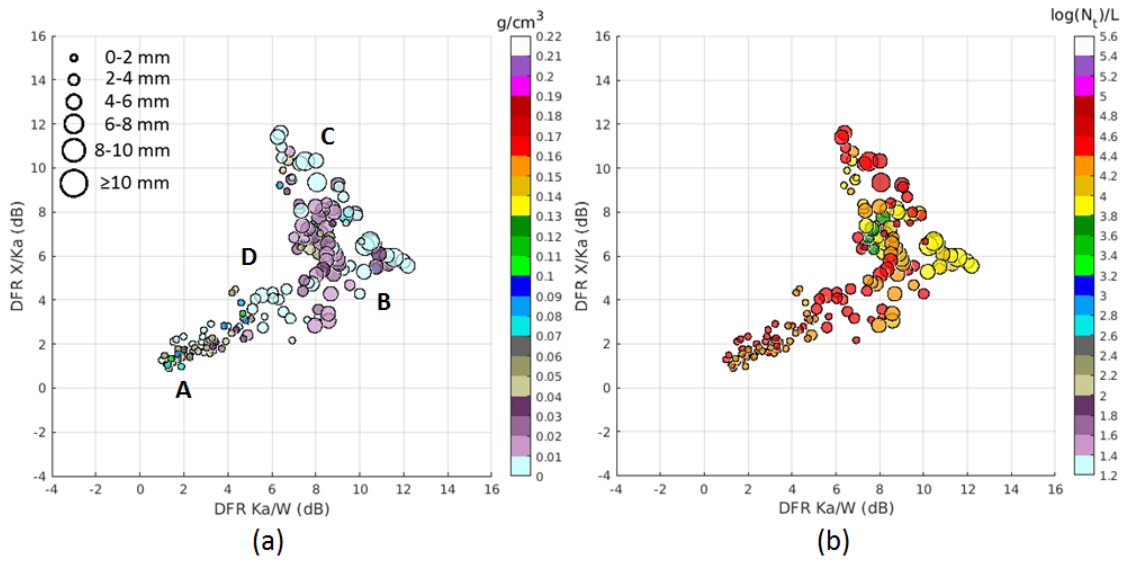

**Figure 20: Similar to Fig. 14 but for flight segment 20:05 UTC – 20:28 UTC.**





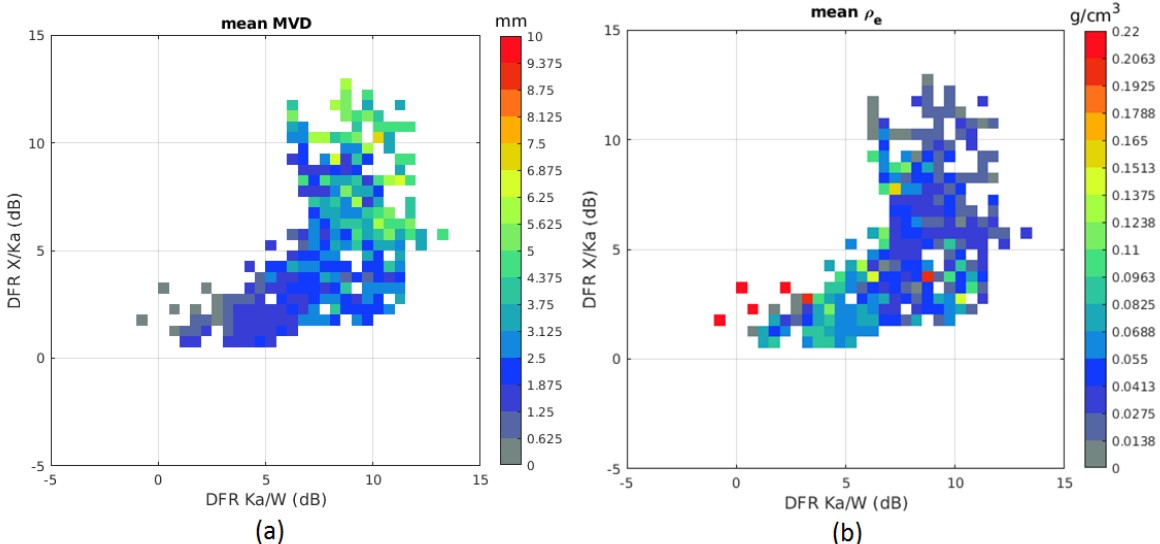

**Figure 21: Mean MVD and $\rho_e$ calculated from all the data points analysed in three study cases in the flight on November 22nd, 2018. The size of the DFR grid is 0.5 dB in both axes.**

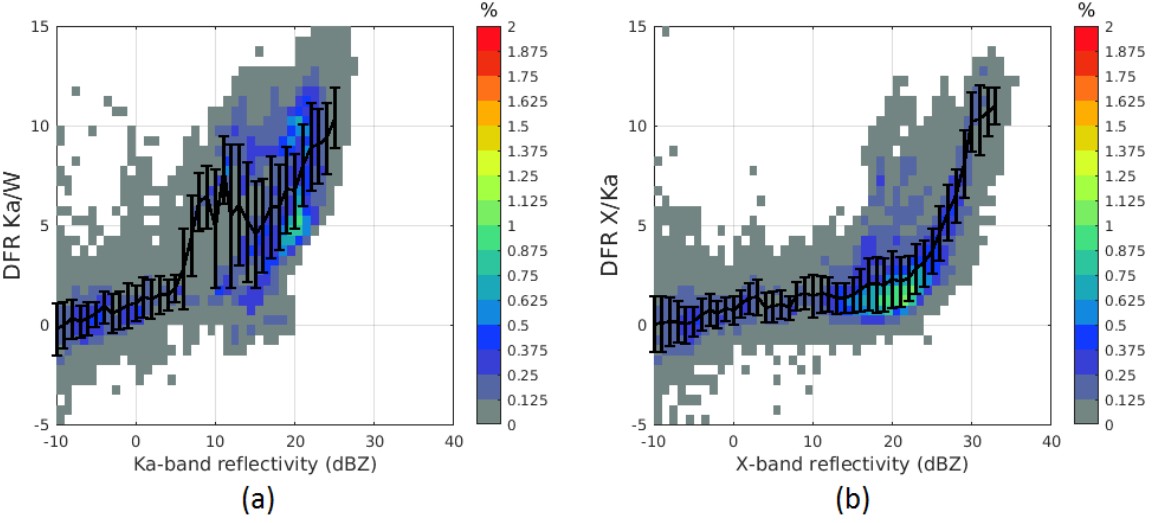

**Figure 22: Occurrence density plot of Ka-band reflectivity vs. DFR Ka/W (a) and X-band reflectivity vs. DFR X/Ka (b). Data are from nadir antennas in the 22 November 2018 flight (Fig. 9) with over 23500 data points.**

## 5 Summary and discussion

The X-Ka-W-band airborne radar observations and almost perfectly co-located in situ microphysical measurements collected during the RadSnowExp project provides unprecedented dataset for radar multi-frequency study of snow/ice clouds. The dataset

includes more than 12 hours of data with more than 3.4 hours in non-Rayleigh regions for at least one of the radar frequencies.



The potential of this dataset is illustrated here using one flight data during an Arctic storm that covers a wide range of snow habits from low pristine ice crystals, low density aggregates to heavily rimed particles with maximum size exceeding 10 mm. The triple-frequency signals (DFRs) in three study cases are observed as large as 12 dB, and they appear to be determined by non-Rayleigh effects only thanks to the close range measurements and additional processing which improve radar volume matching. The study cases were observed in a relatively large temperature range between - 40 and -10 ℃ and at different flight altitudes.

In this work, we focus on finding the relationships between ice particle properties and radar triple-frequency signatures and their potential for developing quantitative retrievals of fundamental ice cloud microphysics. We also provides brief discussions on some measurement aspects (DFR variability and radar sensitivity) which might affect the triple-frequency radar applications.

Preliminary results confirm the main findings of previous modelling works with radar dual frequency ratios (DFRs) moving within different zones of the DFR plane (Leinonen et al. (2012), Kulie et al. (2014)). We find that the size of the crystals has a measurable effect on the triple-frequency signals. The particle diameter increases further from the origin of the DFR plane, with increasing DFR values corresponding to increasing MVD. The signal of the DFR X/Ka and the DFR Ka/W pairs respond to different particle size ranges, with more linear responses for MVD ranges of 2-10 mm and 1-5 mm, respectively. However, saturation of DFR Ka/W for large aggregates can produce crossovers between DFR Ka/W and DFR X/Ka. Reversely, the strong connection between the particle size and the triple-frequency radar signature suggests that the data could be directly used for quantitative retrieval of particle size using measured DFR Ka/W and DFR X/Ka without prior knowledge of the overall cloud composition. A first attempt is shown in Fig. 22a where all data points from three cases of the flight on November 28 are used to estimates MVD. In a similar way, effective bulk density of all data points can be averaged and mapped to the DFR plane (Fig. 22b). We find that, in general, effective bulk density of ice particles decreases as DFR X/Ka decreases and DFR Ka/W increases ($\rho_e$ rotation feature) which in a good agreement with findings in other airborne datasets (Chase et al., 2018). These results look promising but the estimation errors could be high because different combinations of ice particles within the radar volume can produce similar triple-frequency signatures. Future improvement could be obtained by using more data points and a large set of scattering computations; a more quantitative analysis based on a Bayesian retrieval scheme is the topic of a companion study (Mroz et al., 2021b)..

With the high resolution grayscale imagery of small cloud drops from CPI probe, we are able to identify signatures of different types of rimed particles. Regions with DFR Ka/W between [3-12] dB and DFR X/Ka between [2-6] dB are often connected to rimed particles with MVD < 6mm (although millimetre aggregates could also fit into this region). However, the shape of PSD also has noticeable effect on the distribution of DFR values (Mason et al., 2019). For the same characteristic size, data points with broader and flatter PSD tends to bend away from the horizontal curve (higher DFR X/Ka and lower DFR Ka/W). This feature was demonstrated in section D of segment 1 or in section B of the segment 2 where we observed rimed particles with MVD <6 mm  but DFR X/Ka > 8 dB and DFR Ka/W in the range of 6-8 dB. The distribution of rimed particles in the DFR plane found in this study spread in a much wider region than the findings in Kneifel et al. (2015). On the other hand, large and low-density aggregates occurred in the region with both DFRs greater than 8 dB.

A multi-frequency system is intended to be useful because different frequencies are complementary (different sensitivities are exploited) and synergistic (non-Rayleigh scattering effects allow better microphysical retrievals, Battaglia et al., 2020b).  If the highest frequency radar is envisaged to provide sensitivity to small particles (e.g. like thoroughly demonstrated by CloudSat) the lower frequencies must cover only the regions where non-Rayleigh effects become tangible.  Although the RadSnowExp data is very limited, it provides observational unique dataset radar sensitivity requirements for monitoring of arctic clouds.   A first clue about where this happens is provided in Fig. 22. In a X-Ka band (and similarly a Ku-Ka band) system the lowest frequency



ideally should reach at least down to 0 dBZ sensitivity to fully cover non-Rayleigh targets (right panel) with the Ka-band system achieving sensitivities much better than that (thus far better than the current GPM-DPR); similarly in a Ka-W system the Ka-band sensitivity should go down to -5 dBZ (left panel). Recent developments in new technologies make these goals at reach (Battaglia et al, 2020b, Kummerow et al., 2020). Alternatively, an increased DFR dynamic range for small ice particles can be

achieved by including observations at frequencies in the G-band (Battaglia et al, 2020b, Lamer et al., 2021).

Closure studies that try to reconcile in situ PSD and IWC with remote sensing radar reflectivities remain challenging due to spatial variability of microphysics and mismatch between in-situ probe sampled volumes and radar backscattering volumes. Possible solutions can be provided by flight-direction forward or backward looking radars or adopting sophisticated phase coding schemes like Quadratic Phase Coding (Mead and Pazmany, 2019) to significantly reduce the blind zone close to the radar or

multiple aircrafts coordinated flights.

*Acknowledgments.* This work is supported by the ESA RadSnowExp field project (Contract: 4000124359/18/NL/FF/gp), ESA RainCast project (Contract: 4000125959/18/NL/NA) and NRC. Many people from NRC, ECCC and Université du Québec à Montréal (UQAM) contributed for a successful completion of the campaign in a very challenging environment. We thank the

engineering, operation and managerial staffs from NRC (E. Roux, J. Millett, S. Ingram, T. Van Westerop and D. Hoyi) and ECCC (M. Harwood and J. Iwachow) who made the project possible by working long hours during instrument integration and field operations. The authors also would like to acknowledge the contributions of the ECCC science team (A. Korolev, D. Hudak, P. Rodriguez, and Z. Mariani) for their scientific advice and field support, J. P. Blanche and L. Pelletier of UQAM for forecasting support during the campaign.

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
