# Peer review of "Coincident In-situ and Triple-Frequency Radar Airborne Observations in the Arctic"

_Atmospheric Measurement Techniques, 2021_

## Referee Comment (RC2)

**Coincident In-situ and Triple-Frequency Radar Airborne Observations in the Arctic**

**1 General comments**

This is an interesting paper presenting triple-frequency radar observations collected from an Arctic region using X, Ka, and W band radars on board an aircraft, along with microphysical measurements. The NRC Convair-580 aircraft carried the NAWX and KPR radars, along with an array of in-situ and remote sensing sensors. This is different to previous campaigns which usually don't make all the measurements from the same platform.

The authors consider the best way to treat the data, and whether data should be taken from above or below the aircraft. This is done by simulating Z using in-situ PSDs, and comparing to measured Z.

The authors use machine learning to classify CPI imagery into particle habit groups. They also look at imagery from the HVPS3 probe. They then explore where the signal from the various groups is located in triple frequency space, in particular focussing on the relationships between DFRs and MVD and effective bulk density.

Some suggestions are outlined below, there are a lot of points but most of them are minor grammatical corrections. I request that the font size is increased on some of the plots. Moreover, some of the conclusions need to be clarified as the claimed relationship between bulk density and DFR is not obvious to me. Nonetheless, I thoroughly enjoyed reading the manuscript, and look forward to seeing it in it's final form.

**2 Specific comments**

Line 12- I think you need to clarify the opening line of the abstract. If I have understood correctly, this is the first dataset where the airborne radar and microphysics data were collected from instruments on a single aircraft, allowing for very accurate co-location. Other campaigns such as OLYMPEX have collected airborne triple-frequency and microphysics data, but the difference is that the instruments were mounted on 2 different aircraft, so the co-location is less accurate. To me, the opening line of the abstract sounds like you are claiming that nothing similar has ever been done before. Or, is the novelty that the measurements presented here were made in an Arctic region?

Line 39- Le et al, should this be Leroy?

Line 52- I think it would be useful in this paragraph to point out that triple frequency radar measurements have been made using ground based campaigns (e.g. TRIPEx; Dias Neto et al., 2019). The introduction of that paper provides a nice, concise summary of what has been done before, with relevant references. The PICASSO campaign in the UK has also been making ground based triple-frequency measurements along with coincident in-situ aircraft measurements of the microphysics. The co-location is very accurate as the radar dish is steered automatically using the real-time position feed from the aircraft. However, this work has not been published yet.
For example see:

http://blogs.reading.ac.uk/weather-and-climate-at-reading/2019/improving-model-representation
-of-cloud-ice-using-cloud-radar-and-aircraft-observations/
or
https://ams.confex.com/ams/15CLOUD15ATRAD/webprogram/Paper347299.html

Line 65- reference to Fig. 1: Firstly, the figures seem blurry to me. Secondly, in the figure captions here and elsewhere, I would prefer the reference to the subpanel to come before the description, i.e. "(a) Full flight path" instead of "Full flight path (a)". Thirdly, the last thing you mention before referencing Fig. 1 is the altitude, yet this is not shown in the figure as far as I can see. Are the colours in Fig. 1a showing altitude? If so, this needs a key.

Line 71- Reference to Fig. 2 should be Fig. 3. Suggest switching Figs 2 and 3, as Fig. 2 (the triple freq plot) has not been discussed at this point in the manuscript.

Line 95- Should be $k_{f2} - k_{f1}$ in the attenuation bracket

Line 111- do you mean corresponds to leaving the Rayleigh regime as the particle size increases?

Line 112- You point out that the shape and degree of riming of particle models contributes to the differences seen on the triple frequency plot. It is true that there is variability between particle models, but it is worth noting that some of the differences seen here result from the fact that the size range of your particle models is not the same. For example, the sizes of the ice sphere in the ARTS databases go up to $50000\mu m$, while some other particles such as the small column aggregate and GEM cloud ice only have sizes up to 3000-4000$\mu m$. Moreover, the scattering calculations are done using different methods which will make a difference (e.g. DDA vs SSRGA). For example, see Fig. 11 of McCusker et al. (2021) for examples of differences in DFRs calculated using different scattering methods:
https://rmets.onlinelibrary.wiley.com/doi/full/10.1002/qj.3967

Line 132 (Table 1)- need degree symbol for W and Ka beamwidths. Could you clarify what "Sampling resolution" is and how it is relevant to the processing of the data and subsequent analysis/results? It doesn't seem to equal the vertical range resolution, but it is not obvious why there are two numbers listed for each radar.

Line 163- There are 2 items in your bibliography that "Wolde et al., 2019" could be referring to, and the "Nguyen and Wolde (2021)" paper you reference seems to be "Nguyen, Wolde and Pazmany" in the bibliography.

Line 168- It is clear from Fig. 4 that the radar profiles all match up nicely from 700m onwards, with mismatches at closer ranges. A height difference of 700m could be a big difference in microphysics so you want to use radar data that is closer to the aircraft, but in this region the data don't actually match. You select a range of 245m as W band attenuation is minimal, and claim in line 176 that the W band data is not affected by close range biases, and the offsets from the other radars are almost constant for each flight, meaning you can apply a correction for this to obtain an unbiased estimate of DFR.

Could you please address the following 2 related points:
(1) You claim the data don't match in the close-range region due to limitations in the hardware. This is a bit vague, can you explain this is more detail, or consider other hypotheses and their plausibility?
(2) Can you show direct evidence that the difference between the two radars is consistent, and quantify the (hopefully small) variability, so that we know what the uncertainty on DFR is?

Line 180- reference to figure 10d, yet this figure is 7 pages later in the manuscript after many other figures that have not been referenced yet. Is it possible to reorder them to match the order they are referenced?

Line 192-194- rephrase this sentence

Line 206- Are you referring to Figs 13, 16, 19? You could say "The fraction plots are presented in section 4", or something similar, so the reader can find them easily.

Line 224- How did you come up with this value for the estimated value of Nevzorov data?

Line 234- Is $m_{IWC}$ just the ice water content, if so why not call it IWC?

Line 236- I think you should remove the sentence "Both $m_{IWC}$ and V are computed for $1m^3$" and instead clarify units in the previous sentences. For example "$m_{IWC}$ (or IWC?) is the mass of ice inferred from the power dissipated on TWC and LWC sensors of the Nevzorov probe (Korolev et al., 1998), with units of $gm^{-3}$. V is calculated as the sum of the volume of all particles within the PSD, with units of $cm^3 m^{-3}$. Thus $\rho_e$ has units of $gcm^{-3}$"

Line 239- Include the formula, e.g. equation 4 in Brandes et al., 2007. Leroy et al. (2016) describe calculation of MMD, not MVD.

Line 246- "whilst there were almost certainly no similar sampling point" Can you rephrase/clarify this? You could just say something like: "The case study we are looking at is from the 22 November 2018, which we chose because larger values of MVD were more frequent than during the other 2 flights (Fig. 6)."

Line 251 (Fig. 6 caption)- Change "particle diameter from PSD" to "MVD from PSD"

Line 280- The caption of Fig 7 needs more explanation. The black dashed line is just the 1:1 line? What do the solid black line and the bars represent?

Line 294- Again referencing Fig. 10 which appears later in the manuscript, after some figures that have not been referenced at this point. Suggest move Figure 10 to appear earlier in the manuscript, and also increase the font size on the figure as it's not easy to read.

Line 300- I don't think Fig. 8b really shows this, the aircraft altitude is more obvious in Fig. 10a.

Line 320- There are 4 boxes, but I think you only look closer at 1, 2, 4. If this is correct please remove the third box.

Line 335- Might be helpful to label the boxes you are referring to in the figure

Line 349- I think you mean to refer to a different figure here

Line 364- "remarkably mirrors" - I don't think this is very obvious actually

Line 365- You point out that in section C the aircraft sampled heavily rimed dendrites, large aggregates, and reduced drops. From the imagery it looks like unrimed dendrites are also present in section C. From the top particle fraction plot it looks like section C is mainly drops, small ice, and irregular ice, why does this plot not show many rimed particles? Likewise for section D. Perhaps it's just too small an amount to be visible on the scale presented.

Line 369- "In section D, mainly heavily rimed, fractured ice and frozen drops are present" - In the particle fraction plots in Fig. 13 for section D, it looks like "other ice", "pristine" and "dendrites" are mainly present. There is a greater proportion of pristine particles than rimed particles, or am I interpreting the plot incorrectly? When you say fractured ice, are you referring to "other ice" in the groups in Table 2? Also, in the list of particle types for the "Other ice particles" merged group from Table 2, what does "ice" refer to?

Line 372- Now you're saying that in section D the fraction of rimed particles is at its lowest level, when you just said on line 369 that it was mainly heavily rimed particles?

Line 380- Fig. 13 is a nice plot full of important information, but it is difficult to read as the font is blurry, and the HVPS imagery is also blurry. Can you enlarge the font size and the size of the figures? Also label the subpanels and refer to the labels in the text (paragraph starting line 351), rather than referring to the middle right panel, for example. In the MVD and $\rho_e$ figure, the axes colours should match the line colours, otherwise you need a legend.

Line 388- "DFR X/Ka decreases when bulk density increases" - This is not obvious to me, do you mean DFR Ka/W decreases?

Line 392- In the references, sometimes the year is in brackets and sometimes it's not (here and elsewhere in the manuscript). I think the sentence would be easier to read if the references were given at the end of the sentence rather than in the middle. Maybe say "only aggregate models have been shown to produce a "hook signature" as observed in the data displayed here (List references)."

Line 395- Change "heavily riming clouds" to "heavily rimed cloud particles". But also consider whether this is actually the case, e.g. refer to comments for line 369 and 372. Perhaps it should be changed to "only a small proportion of rimed cloud particles".

Line 397- Are sections A and E discussed?

Fig. 14- I really like the way the data is presented in these plots, although some of the pink/purple shades may be difficult to distinguish from each other.

Line 404- Correct attitude to altitude. The black line in Fig. 10a is above 2km, is that the aircraft altitude? You say it's 1.7km here

Line 405- Another issue with figure order, Fig. 16 is referenced before Fig. 15, so suggest switching the order of these 2 figures.

Line 410- Is the fraction of rimed particles and dendrites higher in section B than section C? The combination of orange, yellow, and purple sections in the ice only plot in Fig. 16 seems similar for B and C, if not bigger for C?

Line 412- What do you mean by "increased in land"?

Line 418-420- Your summary of section A needs to be revised. You say MVD is between 2 and 4 mm, but maybe it should be more like $1mm < MVD < 4.5mm$? Also, you say "for larger MVD ($\sim 6mm$)", but looking at the MVD plot (4th panel down at left hand side of Fig. 16), the largest value of MVD in section A is less than 5mm.

Line 422-423- "peaked at $\sim$10 dB...when MVD is greater than 8mm" - There are multiple points on the DFR curve in section B when DFR exceeds 10 dB, while there is only one point when MVD exceeds 8mm. "MVD is greater than 4mm" would be more accurate, or perhaps you could rephrase to something like: "when MVD is large, at times exceeding 8mm"

Line 427- You talk about overlapping data points between sections, but we can't see the boundaries. Is it worth putting boxes around the data points in each section?

Line 428- Do you mean $0.12g/cm^3$?

Line 428-430- You repeat the same thing in 2 sentences here, can you rephrase to be less repetitive?

Line 430- You refer to Fig. 16 left panel - which plot are you referring to?

Fig.16- increase font size and label subpanels, refer to labels in text on page 17. Also do this for Fig. 19 and associated text

Line 449- You say around 2 dB, but around 2-4 dB would be more accurate

Line 453- Should 2-10 be 2-12 dB?

Line 463- Is 2.9km correct? On line 445 you say 2.2km.

Line 464- Specify that the increase in MVD is only at the beginning of the section

Line 470- I'm sure this will be fixed in later versions, but the figure caption of Fig. 18 is not attached to the figure and appears on a different page

Line 499- I don't understand what you mean by the linear response comment, could you expand on this point please?

Line 500- the saturation of DFR Ka/W is discussed in section 4 of Stein et al., 2015. You could perhaps link back to Fig. 2 here, as I don't think you have really made a connection between that figure and your results. Otherwise there is not much need for the figure.

Line 503- Figure 21, not 22. Can you just provide a brief sentence explaining how you created these plots of mean MVD? In line 504 you talk about estimating MVD, but you have plotted "mean MVD".

Line 505- bulk density decreases as X/Ka decreases and KaW increases- this is not obvious to me. Maybe bulk density decreases as X/Ka *increases* and KaW increases? Actually page 5757 of Chase et al. (2018) finds that $\rho_e$ *increases* as KuKa decreases and KaW increases, as in Kneifel et al. (2015), but I can't see that behaviour from your Fig. 21b. Am I missing something?

Line 513- Can you provide references where these DFRs were connected to rimed particles with MVD less than 6mm?

Line 523/524- Rephrase this sentence beginning "Although..."

**3   Technical corrections**

Line 22- there is a double period here, and no period at the end of the abstract. Actually, the abstract seems to be incomplete, unless you mean to say "the analysis shows that *there are* close relationships between the..."

Line 23- remove s from aggregations

Line 30- Change "its data have shown a great potential also for rain estimation and snowfall in particular" to something like: "its data have shown great potential for rainfall and snowfall estimation"

Line 35- performance *is* slightly improved

Line 46- multiple radar frequencies, rather than multi radar frequencies

Line 50- instead of "global distribution of the ice-phase precipitation and, therefore, enhancing our knowl-edge", I would say "global distribution of the ice-phase precipitation, thereby enhancing our knowledge", or "global distribution of the ice-phase precipitation and therefore enhance our knowledge"

Line 53- maybe "capabilities of", instead of "capabilities in"

Line 57- I think the bracket including a sentence along with references separated by commas should be tidied up. One suggestion: "...cloud microphysical data. For example, the OLYMPEX provides 2.2 hours of in-cloud data with Ku-Ka-W radar data and coincident microphysics (Chase et al., 2018; Tridon et al., 2019)."

Line 58- remove such

Line 59- remove comma after low-intensity

Line 64/65- suggested edit: covering a large geographical region and wide range of microphysical conditions

Line 71-75- You are describing what the dataset features, so the points need to relate to this. For example the first point should say "triple-frequency radar data" and the second point should say "data from state-of-the-art in-situ sensors". The second point should end with a semicolon rather than a period. In the third point there is an s at the end of atmospheric.

Line 79- Remove "on"

Line 88/89- Suggest editing this sentence to something like "The large variability in ice crystal properties such as density, size, and shape makes the interpretation of..."

Line 97- Should be "due to non-Rayleigh *effects*"

Line 102- Remove s from attenuations

Line 109- should be "*particle* scattering properties", not "*particles*"

Line 119- "via *the* discrete dipole approximation". $\mu =$ hasn't come out as a symbol. Need to say what $\mu$ is.

Line 130- provide *the* highest level

Line 152- Rephrase, suggestion: "Secondly, data from the three radars is..."

Line 153- Change "mapped into" to "mapped onto"

Line 168- change "within 700m from" to "within 700m of".

Line 173- Remove s from snow

Line 195- "For this work, *the* cloud particle size distribution..."

Line 198- remove space before period

Line 201- should be "determination", remove s from determinations

Line 207 (Table 2)- remove comma after plates

Line 210- "*The* particle size detection range"

Line 219- Korolev et al., 1998- period should be after al, not after et

Line 226- calculating *the* total volume within *the* PSD

Line 227- Do you mean "and *aid* interpretation of parameters"?

Line 230- *the* measured PSD*s* (You are not just using one singular PSD, right?)

Line 232- Heymsfield year, 2004? Put a colon after defined as (rather than a comma)

Line 258- "In *the literature*, collocating radar and and in situ data is often *achieved*"

Line 259- finding *the* nearest airborne radar data points to the in situ *measurements*

Line 264- *on* the same platform

Line 265- *The* temporal sampling rate. Here and on line 269- I don't think there is a section 3.1a and 3.1b?

Line 267- I don't think decimated is the right word, do you mean the radar data is degraded?

Line 271- Remove comma after first

Line 277- *in* the two directions

Line 287- *the* equivalent reflectivity factor

Line 288- Rayleigh-Gan*s*

Line 289- Sometimes you write "X-band" and sometimes "X band". Please keep this consistent.

Line 290- scattering effect*s*

Line 303- *and* densely rimed particles

Line 304- Replace Figure with Fig.

Line 312- (Fig. 9 caption) temperature was in *the* range

Line 330- Reference to non-existent section 3b, do you mean 3.3.2?

Line 338- (Fig. 11 caption) replace PDFs with pdfs

Line 340- temperature*s*

Line 344- do you mean "based *on*"?

Line 344- Fig. 12 show*s*

Line 351- remove ' from panels

Line 366- change "it is worth to note" to "it is worth noting"

Line 368- rephrase "an increase in dendrites portion". Do you mean "an increase in the proportion of dendritic ice habits"?

Line 389- Change "numbers of concentration are" to "the number concentration is"

Line 390- Change "the section B and C data placement" to something like "the location of data from section B and C in the triple-frequency plane"

Line 394- attribute*d* to

Line 413- Rimed *particles* and dendrites

Line 419- Move "respectively" to the previous line, after $\sim 6\ dB$.

Line 452- millimetric needs another l. Also, should it be 2mm rather than 3mm?

Line 455- Change 6000 microns to 6mm for consistency with previous units.

Line 457- Change DRF to DFR

Line 459- change boarder to broader. I think you mean to refer to Fig. 18? Change affect to affects

Line 460- Remove "continually" as the increase is for a short period of time and is not continuous over the remainder of the section

Line 461- Change aggregate to aggregates

Line 462- Remove "that"

Line 483- In the abstract and conclusion you say co-located, but in the main text you say collocated.

Line 484- Suggest rephrasing to something like: "provides *an* unprecedented dataset for studying multi-frequency radar signatures of snow/ice clouds"

Line 485- Perhaps rephrase to something like: "...3.4 hours when the scattering was non-Rayleigh for at least one of the radar frequencies"

Line 486- Rephrase to something like: "illustrated here using data collected from one flight during an Arctic storm..."

Line 487- remove "low" before pristine

Line 488- The dual frequency ratios (DFRs) are as large as 12 dB

Line 488- do you mean "appear to be *dominated* by" ?

Line 493- Remove s from provides

Line 495- If you take my suggestion for line 488, you don't need to spell out DFRs here

Line 504- Remove s from estimates

Line 510- double period

Line 514- the shape of *the* PSD also has noticeable effect*s*

Line 528- Change "at reach" to "within reach"

---

## Author Comment (AC2)

**Responses to the AMT-2021-148 Manuscript RC1 comments**

The paper presents the measurement results from the airborne RadSnowExp campaign which offers near-simultaneous and coincident triple-frequency radar observation and in-situ (in-cloud) ice particle characterization. The observational setup is rather unique as it basically shifts what is commonly done on the ground (e.g. BAECC campaign in Finland, von Lerber et al 2017) on an airplane allowing for direct in-cloud particle imaging, but posing new challenges regarding the observational constraint given by the airborne platform.

The analysis of the dataset focuses on the connection between triple-frequency radar signatures at the X-Ka-W band and particle properties which is a very relevant subject for snow microphysical studies. The study confirms the existence of such connection as it is predicted by various modeling studies which are presented in the paper introduction. The results of the study are supporting the idea of using multifrequency radars for microphysical retrievals.

Given the significance of the dataset presented I think the paper constitutes a valuable contribution to AMT. However, I have a few major comments that I suggest to be addressed before the paper is published.

We would like to thank the referee for these very helpful and constructive comments. We have significantly revised most of the figures to improve their appearance and the overall presentation of the manuscript to enhance its clarity and readability. We have also addressed the various deficiencies pointed out by the referee throughout the paper. Please find our detailed responses to your comments as follows.

1) Figure quality. I do not think that the presentation quality is sufficient for a final publication. Many figures are very hard to evaluate due to the fact that they are quite compressed in terms of the range of values. Also, text and labels are often hardly visible. Some significant work must be put on the figure quality.

The following are just suggestions connected to the aforementioned "readability" point, but it is up to the authors to take it or not. The number and size of the figures are significant, perhaps some work can be done also in this direction to rationalize the figure-load and

facilitate the reading. As an example, Figures 12, 15, and 18 occupy an area comparable to the one occupied by Figure 13 and not equivalently discussed. Perhaps they could be probably be accommodated as subpanels of their respective "event-dashboards" (figures 13, 16, and 19) making this a complete overview of the measurements.

Figure 6 can be combined with Figure 5 giving a single general overview of the flight path and atmospheric conditions which connects well to the description given in the text.

Finally, I see a little relevance of figures 1 and 6 which are not necessary for the paper and can be moved to supplementary material or even left out. The study focuses on the measurements taken during the flight of 22 November and should only present data from that flight in my opinion. I basically had problems during the reading in following the various hierarchical groups: campaign->flights->segments->sections(A, B, C ...).

We completely agree with the referee on this comment. In the revised manuscript, we have increased the font size in the figures and rearranged the panels in Figs 13, 16, and 19 to improve the figure quality and readability (see the response in detailed comment below). We have considered accommodating Fig. 12 as a subpanel of Fig. 13 (similar to Figs. 15 and 16; 18 and 19) but the resultant figure became too "busy". We have also removed Fig 1, 5, and 6 as they are not necessary for the paper. With this major modification we believe the revised manuscript is now easier to read.

2) Data availability. I did not find the data availability section. There are occasions where the paper specifically states the importance of the presented dataset which makes the data availability not only highly recommended, but quite essential to deliver the value of the study to the scientific community.

During the submission process, we entered this information to the Data Availability section on AMT "The data used in this analysis are stored at the National Research Council database. Please contact the lead author for access to the data." Our plan is to publish the whole dataset when all the flight data are quality controlled accompanied by a paper to ESSD describing the details of all flights in this project.

3) Scope and Uncertainties. The abstract (and in part also the summary) states that there is a "close relationship" between triple-frequency and particles' bulk density, level of riming, aggregation, and characteristic size of the PSD.

The degrees of aggregation and riming are not evaluated if not only qualitatively, but I do not understand from this paper how to use DFR to make a quantitative estimation of aggregation and riming degree.

We have revised the text in the abstract and conclusion sections to remove this confusion on the quantitative estimation of aggregation and riming degree. The accompanying paper (Mroz et al., 2021) presents nice results on the estimation of level of riming using the same dataset mentioned in this manuscript.

Regarding bulk density, I do not see such close relation. Judging from figures 14, 17, and 20 it seems that bulk density is connected to mean size but can take various values at the same DFR range. Looking at Fig 21b it seems that high-density values are found for small DFR and on both left and right sides of the histogram. The "rotation" feature in the triple-frequency plot is not really evident. The range of bulk density values is very limited and skewed towards low densities which suggests a general problem in estimating this quantity. This also suggests that higher density values are found at the borders of the histograms due to problems of statistical representativeness (rare values are found in small samples). Density values in Fig 21 seem to correlate mostly with MVD rather than DFR.

We agree with this referee's observations and address his remarks in the section below.

Regarding MVD I think that a correlation with DFRs is clear. However, Fig 21a only shows the mean MVD for a combination of DFRs and does not show other important quantities such as the variance of MVD which I believe is essential for the retrieval study of Mroz (2021).

We have added the variance plot for MVD and $\rho_e$ estimation to Fig. 21.

Detailed Points
Line 115. Figure 2 - it is very difficult to connect the curves to the legend symbols. Perhaps enlarge the legend fonts or group the legend labels in different blocks according to their respective main group (already color-coded). Also, it is not totally clear to me how this is used in the study. If it is only for illustrative purposes or it is actually an attempt to connect with microphysical properties?

For example, It would be nice to connect the triple-frequency characteristic of these modeled particles with microphysical quantities as they are defined in section 3.2 lines 230-240. What are the MVD and bulk density of these modeled particles? How do they compare with the mean values measured for the same DFRs?

Fig. 2 has been introduced here only for illustrative purposes. A more quantitative analysis that discusses the values expected for the microphysical properties (Dm and density) in correspondence to a given pair of DFR values (DFR Ka-W, DFR X-Ka) is provided in a companion paper (Mroz et al. 2021, in particular see Fig.2).

The figure has been modified by increasing the symbol size and the legend fonts and identifying for some of the symbols the corresponding MDV values.
Line 161-168 If the Ka and W band radars are absolutely calibrated, their return for Rayleigh ice particles should be around 1.2 dB and not 0. This is because of the frequency-dependent difference between the dielectric factor K for ice and water. The radars cannot be simultaneously calibrated in an absolute sense and have DFR=0 for small ice particles. Please clarify the calibration procedure.
This point is also discussed in Dias Neto et al. 2019 and Ori et al 2020

In the calibration constants, we use dielectric factor (|Kw|^2) of 0.93, 0.88, and 0.7 for X, Ka, and W-band at T=0° C, respectively. The cross calibration between frequencies is done at the region of small ice particle sizes (e.g. near cloud top at the beginning (19:07 – 19:32 UTC) of the 22 Nov flight). For small ice particles (median size less than 300 um), according to Matrosov et al. (1993), differences between $Z_X$ and $Z_{Ka}$ should be negligible, and differences between $Z_{Ka}$ and $Z_W$ is small, $Z_{Ka} - Z_W \sim 0.2 \, dB$. We have updated the radar data and re-generated all of the related figures. The text in this paragraph has been revised for clarity.

https://psl.noaa.gov/people/sergey.matrosov/1993%20-%20jgr.pdf

Line 185 Figure 4 It is very difficult to evaluate a bias of 0.8 dB on a small scale that spans over 100 dBZ. Considering the objective of the figure I would cut it between -15 and 10 dBZ focusing on the upper part only.

We have removed panel (a) and added a "zoom-in" plot of the upper part to more clearly show the mismatch in reflectivities at the close range. In the left panel, we prefer to retain the profile reaching to the ground to show the good alignment of the data with the ground return as a reference.

Line 236 It would be nice to include a formula also here like it is done for the other quantities. Usually it is define as,

$$\int_0^{MVD} V(D) N(D) dD = \int_{MVD}^{\inf} V(D)N(D)dD$$

where V(D) is the volume as a function of size.
Since the video disdrometer cannot measure the volume of snowflakes (which is also ill-defined considering that snowflakes' shapes are irregular) it is better to say also how volume is calculated here. Is it still assumed to be a spheroid with a 0.6 aspect ratio? The given citation seems inappropriate to me. Leroy (2016) describes a methodology to calculate Median Mass Diameter (MMD) and it is not clear how this is connected to MVD.

We agree with the referee on this comment. In the revision, we added a formula for MVD as suggested. In the calculation of V(D), each particle is approximated as an oblate spheroid with an aspect ratio of 0.6 similar to that in the calculation of effective bulk density. We added text to clarify this point.
We also thank the referee for pointing out an error in the citation. Since the formula for MVD is added, we removed the reference for MVD.

Finally, the statement "This is the characteristic diameter that contributes most to cloud liquid water or mass" is confusing and incorrect to me. By definition, the size contributing the most to the mass should be the one that maximizes the function m(D)N(D)dD (i.e. the mode of the mass distribution). Even considering the volume equivalent to mass (by

assuming constant density) stating that MVD is the size that contributes the most to the total volume would be again incorrect. The mode and the median value of distribution are in general diverse, this is especially true for multimodal distributions such it is the case in the presented case studies.

We thank the referee for pointing out this error. The text was used in an older version of the manuscript for other parameter and should be changed. In the revision, we replaced it with "The in situ derived MVD will be used to evaluate the relationship between the characteristic size of the PSD and the DFRs (Kneifel et al., 2015). "

Lines 291-294. I am not really sure if I can understand these sentences. First, a 10-minute running window corresponds to roughly 6 km considering the average ground speed. Is homogeneity important for this thresholding technique? How is the threshold of 0.6 identified? What do the authors mean by "accurate analysis"?
I guess that a good correlation is one easy indicator that the authors can use in order to connect measurements on-board of the aircraft and apart from it, but I wonder if this analysis could be biased by the characteristics of the measurements. As an example: If the cloud field analyzed is very homogeneous both measurements would result in a signal mostly dominated by random noise and thus even if the two signals are connected in reality the correlation coefficient would be close to 0.

We tried several ways to determine what cloud segments and which direction (up or down) would be used in the study cases and found (empirically) that the simple correlation method works best. The reason we chose a 10-minute (or 60 km) window is exactly as pointed out by the referee: to avoid the case where the cloud field is homogeneous. In the environment we flew (this Artic storm), the likelihood of the cloud being homogeneous over a 60 km scale is utterly negligible. On the other hand, if a longer window is used, the results will be smoothed out, possibly leading to an inaccurate selection. We mention in passing that it is well known (e.g. Guillaume et al. 2018 and references therein) that atmospheric fields generally exhibit strong spatial scale dependence; this complicates the matter of computing their unbiased averages and other statistical properties (e.g. Selvam 2009 and references therein).

The figure below shows the histogram of the nadir correlation coefficient. The histogram show a main mode with correlation coefficient greater than 0.6. Hence, we selected a threshold of 0.6 for the decision of a good match between the in situ measurements and the radar data.

[Figure]

Line 320 Figure 10 This figure is not readable. I suggest the authors make much better use of the page real estate; increasing the vertical size of the figure, allowing for a better evaluation of the various curves, and significantly enlarging the font sizes.
We thank the referee for this suggestion. We have increased the vertical size of the figure and the font size of figure 10 to improve the readability.

Line 338 Figure 11 Enlarge axes font size of the legend.
Correct as suggested.

Line 346 Is it possible that the 30um peak is due to the shattering of ice particles at the probes? Shattering is not discussed in the text. The reference list includes Lawson (2011) but that reference is not present in the text (Line 600).

To minimize hydrometeor shattering the probes are equipped with anti-shattering tips (Korolev et al., 2013) as it might be possible to see in Fig. 3b. Therefore, the shattering

events are expected to be relatively rare and wouldn't be visible on an averaged size distribution such as the one in Fig. 12. We have removed Lawson 2011 from references and added Korolev et al., 2013. We also amended the sentence in the manuscript for clarity: "The probes are equipped with anti-shattering tips (see Korolev et al., 2013 for details) and were calibrated with glass beads and a spinning chopper before the campaign and re-evaluated in NRC's altitude icing wind tunnel after the campaign."

Field, P. R., A. J. Heymsfield, and A. Bansemer. " Shattering and Particle Interarrival Times Measured by Optical Array Probes in Ice Clouds", Journal of Atmospheric and Oceanic Technology 23, 10, 1357-1371,  https://doi.org/10.1175/JTECH1922.1, 2006.

Korolev, Alexei, Edward Emery, and Kirk Creelman. " Modification and Tests of Particle Probe Tips to Mitigate Effects of Ice Shattering", Journal of Atmospheric and Oceanic Technology 30, 4, 690-708, https://doi.org/10.1175/JTECH-D-12-00142.1, 2013.

Line 380 Figure 13 (the same applies to figure 16 and 19). I like these overview plots, but the Figures are barely readable at maximum magnification on a screen. Also, the subpanels are not labeled and it is difficult to follow the discussion on them. I suggest significantly increase the size of the figures. An idea to make better use of the page surface could be to put all time-plots on the left column sharing the same x-time axis and the ABCDE-sections classification. The left column could take up to 2/3 or even 3/4 of the figure width. Then, the snow images could be arranged on the right column. Also, I suggest reducing the number of ice images including only a few significant ones.
We thank the referee for this suggestion. In the revised manuscript, we implemented this idea and it greatly improves the presentation of the figures.

Line 480 Figure 22. What are the black lines?
The black lines present data means and error bars of one standard deviation of the DFRs. We added a sentence for clarity.

Line 493 It is not clear to me where to find the relationships between ice particle properties and triple-frequency signature in this study. The paper presents a qualitative assessment of relations among these quantities

We agree with the referee on this comment. We have revised the sentence. It now reads "In this work, we focus on evaluating understanding the relationships between …"

Lines 503 and 505. I guess here it refers to Figure 21 and not 22
Thanks the referee for pointing out this error. It is now corrected.

Line 505 I actually see a very little sensitivity of estimated bulk density to triple frequency. From Fig 21b I do not see a transition from more reddish colors to blue/grey while "rotating" counterclockwise in the triple-frequency plot. Can the authors illustrate better this point?

We have reprocessed the Nevrozov data and recalculated the bulk density. The new data are slightly cleaner. We have also changed the colormap and display scale to better illustrate this observation.

Line 506 I saw that the "rotation feature" was much better illustrated in the first version of the manuscript uploaded. And I think that the text got it the other way around, or? A decrease in effective bulk density is expected when DFR X-Ka increases (counterclockwise rotation). For high values of DFR KaW and the low value of DFR X-Ka, we expect denser particles.

We thank the referee for pointing out this error. It should read "effective bulk density of ice particles increases as DFR X/Ka decreases and DFR Ka/W increases". As mentioned above, figure 21 is revised and better illustrates the "rotation feature"

Minor Points
Line 17 Please introduce the CPI acronym
CPI acronym is added.

Line 18 DFR acronym is introduced later at line 21
This mistake is now corrected.

Line 22 Double period ..

Correction has been made.

Line 22 I guess the phrase was intended without the word "that", but I would suggest rephrasing it anyway to make it easier to understand.

The sentence has been rephrased in the revision.

Line 56 Mismatched parenthesis ))

Thank you for pointing out this error. It is now corrected.

Line 95 I think there is a sign problem in the attenuation component of Eq 1. Assuming the attenuation to be semi defined positive such that the measured reflectivity $Z=Ze-A$ then $DFR = Z1-Z2 - (A1-A2) = Z1-Z2 + (A2-A1)$ [Lehrmitte 1990, Tridon 2020]

We apologize for the error. It has been corrected.

Line 232 Missing year in Heymsfield et al.

The missing year has been added.

Line 288 misspelled Gans?

We agree with the referee. It has been corrected.

Line 350 Figure 12. The caption refers to panels (a) and (b) but the figure panels are not labeled. The same applies to Figures 15 and 18.

We apologize for the error. The labels have been added.

Line 351 Text refers to left/right panels, but it is better to use panels labels (a) (b) according to AMT guidelines

We thank the referee for this comment. Figure 13 (also Figs. 16 and 19) is revised and panel labels have been added. We've also revised the text to reflect the change.

Line 436 Figure 15. I guess the caption refers to sections selected from figure 16

The figure number has been corrected.

Line 497 The MEAN particle diameter.

We agree with the referee. It now reads "The mean particle diameter increases …"

Line 546-550 I think that usually, 2020a comes before 2020b in the reference list.

We thank the referee for pointing this error. The reference order has been fixed and the corresponding reference in the manuscript has been changed.

---

## Author Comment (AC3)

**Coincident In-situ and Triple-Frequency Radar Airborne Observations in the Arctic**

**1 General comments**

This is an interesting paper presenting triple-frequency radar observations collected from an Arctic region using X, Ka, and W band radars on board an aircraft, along with microphysical measurements. The NRC Convair-580 aircraft carried the NAWX and KPR radars, along with an array of in-situ and remote sensing sensors. This is different to previous campaigns which usually don't make all the measurements from the same platform.

The authors consider the best way to treat the data, and whether data should be taken from above or below the aircraft. This is done by simulating Z using in-situ PSDs, and comparing to measured Z.

The authors use machine learning to classify CPI imagery into particle habit groups. They also look at imagery from the HVPS3 probe. They then explore where the signal from the various groups is located in triple frequency space, in particular focusing on the relationships between DFRs and MVD and effective bulk density.

Some suggestions are outlined below, there are a lot of points but most of them are minor grammatical corrections. I request that the font size is increased on some of the plots. Moreover, some of the conclusions need to be clarified as the claimed relationship between bulk density and DFR is not obvious to me. Nonetheless, I thoroughly enjoyed reading the manuscript, and look forward to seeing it in its final form.

We would like to thank the reviewer for many detailed comments and great suggestions which helped improve the manuscript greatly. We have made significant changes in the manuscript and added new figures. We have also reprocessed the Nevrozov data which is now slightly cleaner and recalculated of the bulk density. The discussion of the relationship between bulk density and DFRs has been revised.

We hope these changes make the paper easier to follow.

**2 Specific comments**

Line 12- I think you need to clarify the opening line of the abstract. If I have understood correctly, this is the first dataset where the airborne radar and microphysics data were collected from instruments on a single aircraft, allowing for very accurate co-location. Other campaigns such as OLYMPEX have collected airborne triple-frequency and microphysics data, but the difference is that the instruments were

mounted on 2 different aircraft, so the co-location is less accurate. To me, the opening line of the abstract sounds like you are claiming that nothing similar has ever been done before. Or, is the novelty that the measurements presented here were made in an Arctic region?

We agree with the referee on this comment. We have revised the text in the abstract to make this point clear.

Line 39- Le et al, should this be Leroy?

The reference is Le et al. (2016). We apologized for the missing reference. It is now added to the manuscript.

Le, M. and Chandrasekar, V.: Enhancement of dual-frequency classification module for GPM DPR, in: 2016 IEEE International Geoscience and Remote Sensing Symposium (IGARSS), IGARSS 2016 - 2016 IEEE International Geoscience and Remote Sensing Symposium, https://doi.org/10.1109/igarss.2016.7729550, 2016.

Line 52- I think it would be useful in this paragraph to point out that triple frequency radar measurements have been made using ground based campaigns (e.g. TRIPEx; Dias Neto et al., 2019). The introduction of that paper provides a nice, concise summary of what has been done before, with relevant references. The PICASSO campaign in the UK has also been making ground based triple-frequency measurements along with coincident in-situ aircraft measurements of the microphysics. The co-location is very accurate as the radar dish is steered automatically using the real-time position feed from the aircraft. However, this work has not been published yet. For example see:

1) http://blogs.reading.ac.uk/weather-and-climate-at-reading/2019/improving-model-representation

-of-cloud-ice-using-cloud-radar-and-aircraft-observations/

Or

2) https://ams.confex.com/ams/15CLOUD15ATRAD/webprogram/Paper347299.html

We thank the referee for providing those references. We have mentioned those ground based campaigns to the revised manuscript.

Line 65- reference to Fig. 1: Firstly, the figures seem blurry to me. Secondly, in the figure captions here and elsewhere, I would prefer the reference to the subpanel to come before the description, i.e. "(a) Full flight path" instead of "Full flight path (a)". Thirdly, the last thing you mention before referencing Fig. 1 is the altitude, yet this is not shown in the figure as far as I can see. Are the colours in Fig. 1a showing altitude? If so, this needs a key.

We received comments from other reviewers on this figure and figure 5 and 6. Because they are not necessary for the paper, in the revision, we have removed them and amended the text.

Line 71- Reference to Fig. 2 should be Fig. 3. Suggest switching Figs 2 and 3, as Fig. 2 (the triple freq plot) has not been discussed at this point in the manuscript.

We thank the referee for this comment. The error has been corrected. We have moved Fig, 3 to the introduction section.

Line 95- Should be kf2 − kf1 in the attenuation bracket.

We agree with the referee. Correction has been done.

Line 111- do you mean corresponds to leaving the Rayleigh regime as the particle size increases?

Yes, it is correct. We have modified the sentence for clarity.

Line 112- You point out that the shape and degree of riming of particle models contributes to the differences seen on the triple frequency plot. It is true that there is variability between particle models, but it is worth noting that some of the differences seen here result from the fact that the size range of your particle models is not the same. For example, the sizes of the ice sphere in the ARTS databases go up to 50000µm, while some other particles such as the small column aggregate and GEM cloud ice only have sizes up to 3000-4000µm. Moreover, the scattering calculations are done using different methods which will make a difference (e.g. DDA vs SSRGA). For example, see Fig. 11 of McCusker et al. (2021) for examples of differences in DFRs calculated using different scattering methods:

https://rmets.onlinelibrary.wiley.com/doi/full/10.1002/qj.3967

Yes, we agree with the referee's comment. Indeed the idea of plotting different scattering models from a variety of database was to account for the variability of different methods (as highlighted by e.g. Leinonen et al. 2017; McCusker et al, 2020). Different habits of course have different size range. With the new figure, where we have highlighted characteristic values of PSD characteristic size, it should be clear that certain habits cannot indeed support large sizes (for instance small column aggregate or GEM cloud ice do not have Dm at 4 and 6 mm).

Leinonen, J., Kneifel, S. and Hogan, R.J. (2017) Evaluation of the Rayleigh–Gans approximation for microwave scattering by rimed snowflakes. Quarterly Journal of the Royal Meteorological Society, 144(S1), 77– 88.

McCusker, K, Westbrook, CD., Tyynelä, J. An accurate and computationally cheap microwave scattering method for ice aggregates: the Independent Monomer Approximation. Q J R Meteorol Soc. 2021; 147: 1202– 1224. https://doi.org/10.1002/qj.3967

Line 132 (Table 1)- need degree symbol for W and Ka beamwidths. Could you clarify what "Sampling resolution" is and how it is relevant to the processing of the data and subsequent analysis/results? It doesn't seem to equal the vertical range resolution, but it is not obvious why there are two numbers listed for each radar.

The sampling resolution (or range gate sapcing) is defined as $\tau_s = cT_s/2$ where Ts is the data sampling rate. In the cases of the X- and Ka-band radars, two different options of Ts was used.

Line 163- There are 2 items in your bibliography that "Wolde et al., 2019" could be referring to, and the "Nguyen and Wolde (2021)" paper you reference seems to be "Nguyen, Wolde and Pazmany" in the bibliography.

We apologize for this confusion. In the revised manuscript, we have changed the reference to:

Nguyen, C., M. Wolde, and A. Pazmany, 2019a: The NRC W- and X-band Airborne Radar Systems: Calibration and Signal Processing, 39th Conf. on Radar Meteorology, Iraka, Nara, Japan, Amer. Meteor. Soc.

Line 168- It is clear from Fig. 4 that the radar profiles all match up nicely from 700m onwards, with mismatches at closer ranges. A height difference of 700m could be a big difference in microphysics so you want to use radar data that is closer to the aircraft, but in this region the data don't actually match. You select a range of 245m as W band attenuation is minimal, and claim in line 176 that the W band data is not affected by close range biases, and the offsets from the other radars are almost constant for each flight, meaning you can apply a correction for this to obtain an unbiased estimate of DFR.

Could you please address the following 2 related points:

(1) You claim the data don't match in the close-range region due to limitations in the hardware. This is a bit vague, can you explain this is more detail, or consider other hypotheses and their plausibility?

In the time interval between the instant when the transmitter stops transmitting and the instant when the receiver reach to its steady state, the receiver gain can vary greatly. An example below demonstrates this phenomenon. It shows the system noise of the X-band radar as a function of range (or time). The X-band receiver gain should be stable within 1 dB standard deviation after 1 km. Usable range for the W-band (not shown here as we have to go back to raw data to generate a similar one) is about 200m.

[Figure]

(2) Can you show direct evidence that the difference between the two radars is consistent, and quantify the (hopefully small) variability, so that we know what the uncertainty on DFR is?

In the figure below, the scatter plots of cross-calibrated W, Ka and X reflectivities at 245 m from the nadir antennas for the 22 Nov flight are shown. The data are thresholded by MVD < 300um so that $Z_{Ka} \sim Z_X$ and $(Z_{Ka} - Z_w) \sim 0.2\ dB$ (Matrosov, 1993). The black lines are the mean and error bars (one standard deviation). We found that the standard deviation of the DFRs is 0.8 dB in average. We have included this information and these plots in the revision.

[Figure]

Line 180- reference to figure 10d, yet this figure is 7 pages later in the manuscript after many other figures that have not been referenced yet. Is it possible to reorder them to match the order they are referenced?

We have revised this paragraph and included the scatter plots (above) to make the presentation clear.

Line 192-194- rephrase this sentence

We have rephrased this sentence. It now reads "Bulk liquid water content (LWC) and total water content (TWC), measured simultaneously with a particle imager and spectrometers, were characterized by size distributions, ranging from small cloud droplets to large precipitation hydrometeors."

Line 206- Are you referring to Figs 13, 16, 19? You could say "The fraction plots are presented in section 4", or something similar, so the reader can find them easily.

We thank the referee for this comment. The sentence has been added.

Line 224- How did you come up with this value for the estimated value of Nevzorov data?

An uncertainty of about 0.05 g/m3 is present in the Nevzorov water content data due to uncertainty in the baseline ('dry term'), similar to results shown by (Faber et al 2018). With regard to LWC, in a post-campaign wind tunnel test, the sensor demonstrated standard performance for a Nevzorov probe, similar to (Schwarzenboeck et al 2009).

Faber, S., French, J. R., and Jackson, R.: Laboratory and in-flight evaluation of measurement uncertainties from a commercial Cloud Droplet Probe (CDP), Atmos. Meas. Tech., 11, 3645–3659, https://doi.org/10.5194/amt-11-3645-2018, 2018.

Schwarzenboeck, A., Mioche, G., Armetta, A., Herber, A., & Gayet, J. F. (2009). Response of the Nevzorov hot wire probe in Arctic clouds dominated by very large droplet sizes. Atmospheric Measurement Techniques Discussions, 2(3), 1293-1320.

Line 234- Is mIWC just the ice water content, if so why not call it IWC?

It is correct. We have changed $m_{IWC}$ to IWC.

Line 236- I think you should remove the sentence "Both mIWC and V are computed for 1m3" and instead clarify units in the previous sentences. For example "mIWC (or IWC?) is the mass of ice inferred from the power dissipated on TWC and LWC sensors of the Nevzorov probe (Korolev et al., 1998), with units of gm−3. V is calculated as the sum of the volume of all particles within the PSD, with units of cm3m−3. Thus $\rho_e$ has units of gcm−3"

We thank the referee for this comment. We have changed the text as suggested.

Line 239- Include the formula, e.g. equation 4 in Brandes et al., 2007. Leroy et al. (2016) describe calculation of MMD, not MVD.

We have added a formula for MVD. The error on the reference has been corrected.

Line 246- "whilst there were almost certainly no similar sampling point" Can you rephrase/clarify this? You could just say something like: "The case study we are looking at is from the 22 November 2018,

which we chose because larger values of MVD were more frequent than during the other 2 flights (Fig. 6)."

What we meant by "similar sampling point" is "sampling point in the non-Rayleigh region". We agree with the referee on making this paragraph simpler. In addition, figure 6 is not important for the paper and we have removed it. The paragraph has been amended to reflect the change.

Line 251 (Fig. 6 caption)- Change "particle diameter from PSD" to "MVD from PSD"

In the revised manuscript, figure 6 has been removed.

Line 280- The caption of Fig 7 needs more explanation. The black dashed line is just the 1:1 line? What do the solid black line and the bars represent?

We have added more information to the caption of Fig. 7. The black dashed line is just the 1:1 line and the solid black lines present the mean and error bar of one standard deviation at each DFR bin.

Line 294- Again referencing Fig. 10 which appears later in the manuscript, after some figures that have not been referenced at this point. Suggest move Figure 10 to appear earlier in the manuscript, and also increase the font size on the figure as it's not easy to read.

We agree with the referee on this comment and have fixed the issue with the figure order in the revised manuscript. We have also revised all the figures to improve their readability and presentation.

Line 300- I don't think Fig. 8b really shows this, the aircraft altitude is more obvious in Fig. 10a.

We have updated Fig. 8 with lat and lon lines and the aircraft elevation profile.

Line 320- There are 4 boxes, but I think you only look closer at 1, 2, 4. If this is correct please remove the third box.

Correction has been made as suggested.

Line 335- Might be helpful to label the boxes you are referring to in the figure.

We have numbered the boxes for clarity.

Line 349- I think you mean to refer to a different figure here.

We thank the referee for pointing out this error. It should be referred to Fig. 13. In the revision, we have combined both Fig. 12 and 13 into so the reader would be able follow the discussion easier.

Line 364- "remarkably mirrors" - I don't think this is very obvious actually.

We have changed the sentence to "For example the slight decrease in DFR values around the middle of section B (around 19:51 UTC) resembles the decrease in the relative concentrations of dendrites and rimed particles (Fig. 13, middle right panel)."

Line 365- You point out that in section C the aircraft sampled heavily rimed dendrites, large aggregates, and reduced drops. From the imagery it looks like unrimed dendrites are also present in section C. From the top particle fraction plot it looks like section C is mainly drops, small ice, and irregular ice, why does this plot not show many rimed particles? Likewise for section D. Perhaps it's just too small an amount to be visible on the scale presented.

In section C, small drops, small ice, and irregular ice still dominate but the fraction of rimed dendrite, unrimed dendrites, and large aggregates increased compared to the second half of section B. We have revised the text for more accuracy. "Section C is from sampling of the storm when the aircraft sampled clouds with some heavily rimed dendrites and large aggregates as observed by the HVPS2 probe. The fraction of rimed dendrite, unrimed dendrites, and large aggregates increases and the fraction of small drops decreases compared to the second half of section B."

Line 369- "In section D, mainly heavily rimed, fractured ice and frozen drops are present" - In the particle fraction plots in Fig. 13 for section D, it looks like "other ice", "pristine" and "dendrites" are mainly present. There is a greater proportion of pristine particles than rimed particles, or am I interpreting the plot incorrectly? When you say fractured ice, are you referring to "other ice" in the groups in Table 2? Also, in the list of particle types for the "Other ice particles" merged group from Table 2, what does "ice" refer to?

We agree that the word "mainly" should be removed.

As shown in Table 2, 'other ice' includes ice crystals that do not fall under the other listed groups. Fractured ice crystals can fall into 'other ice' or a different group depending on their look, for example, a large part of a broken dendrite can be classified as dendrite, while a smaller part would fall under 'other ice'. Fractured ice in particular was highlighted here based on the visual review of the CPI imagery, as can be seen in the example in Fig 13.

In Table 2, 'ice' refers to ice crystals that do not fall under other subgroups. 'Tiny ice' refers to crystals whose shape cannot be meaningfully classified because the images are too small (but >= 40 um). Other subgroup names are self-explanatory.

Line 372- Now you're saying that in section D the fraction of rimed particles is at its lowest level, when you just said on line 369 that it was mainly heavily rimed particles?

We thank the referee for this comment. We have removed the word "mainly" and modified the sentence in line 372 to "It also shows a slight increase in the larger sizes whilst the fraction of dendrites and rimed particles drops to its lowest level at the first half of the section when the highest DFR X/Ka occurs".

Line 380- Fig. 13 is a nice plot full of important information, but it is difficult to read as the font is blurry, and the HVPS imagery is also blurry. Can you enlarge the font size and the size of the figures? Also label the subpanels and refer to the labels in the text (paragraph starting line 351), rather than referring to the middle right panel, for example. In the MVD and ρe figure, the axes colours should match the line colours, otherwise you need a legend.

We have rearranged the Fig. 13 panel, increased the font size and labeled all the panels. We hope the modification would facilitate the reading.

Line 388- "DFR X/Ka decreases when bulk density increases" - This is not obvious to me, do you mean DFR Ka/W decreases?

We have reprocessed the Nevrozov data and recalculated the bulk density. The new results are slightly cleaner. We have also changed the colormap and display scale to better illustrate this observation.

Line 392- In the references, sometimes the year is in brackets and sometimes it's not (here and elsewhere in the manuscript). I think the sentence would be easier to read if the references were given at the end of the sentence rather than in the middle. Maybe say "only aggregate models have been shown to produce a "hook signature" as observed in the data displayed here (List references)."

We thank the referee for this suggestion. We have made the changes as suggested.

Line 395- Change "heavily riming clouds" to "heavily rimed cloud particles". But also consider whether this is actually the case, e.g. refer to comments for line 369 and 372. Perhaps it should be changed to "only a small proportion of rimed cloud particles".

We agree with the referee. We have changed the text to "only a small proportion of rimed cloud particles".

Line 397- Are sections A and E discussed?

The discussion for section A and E has been added in the revised manuscript.

Fig. 14- I really like the way the data is presented in these plots, although some of the pink/purple shades may be difficult to distinguish from each other.

We have revised this and other figures, employing a different colormap and reduced display interval for an improved presentation.

Line 404- Correct attitude to altitude. The black line in Fig. 10a is above 2km, is that the aircraft altitude? You say it's 1.7km here

We thank the referee for pointing out this error. It is 2.4 km. The error is now corrected.

Line 405- Another issue with figure order, Fig. 16 is referenced before Fig. 15, so suggest switching the order of these 2 figures.

In the revision, we have merged the two figures into one and checked for the figure order issue.

Line 410- Is the fraction of rimed particles and dendrites higher in section B than section C? The combination of orange, yellow, and purple sections in the ice only plot in Fig. 16 seems similar for B and C, if not bigger for C?

We have corrected this sentence. "In section B, the fraction of rimed particles is the highest.". The rimed particle fraction (purple and yellow) is the highest.

Line 412- What do you mean by "increased in land"?

We meant when the aircraft flew over land. This is not important for the discussion so we have removed the phrase "in land".

Line 418-420- Your summary of section A needs to be revised. You say MVD is between 2 and 4 mm, but maybe it should be more like 1mm < MV D < 4.5mm? Also, you say "for larger MVD (∼ 6mm)", but looking at the MVD plot (4th panel down at left hand side of Fig. 16), the largest value of MVD in section A is less than 5mm.

We apologize for this confusion. We had updated the PSD calculation but some text was not changed. The error is now corrected.

Line 422-423- "peaked at ∼10 dB...when MVD is greater than 8mm" - There are multiple points on the DFR curve in section B when DFR exceeds 10 dB, while there is only one point when MVD exceeds 8mm. "MVD is greater than 4mm" would be more accurate, or perhaps you could rephrase to something like: "when MVD is large, at times exceeding 8mm"

We have fixed this error. The sentence is now read "… at times reached the same level as DFR Ka/W at around 8 dB when MVD is greater than 4 mm."

Line 427- You talk about overlapping data points between sections, but we can't see the boundaries. Is it worth putting boxes around the data points in each section?

In the revised manuscript, we have used different colors for the edge the dots to help identify sections A-C easier.

Line 428- Do you mean 0.12g/cm3?

Yes. The error has been corrected.

Line 428-430- You repeat the same thing in 2 sentences here, can you rephrase to be less repetitive?

We thank the referee for pointing this out. We have fixed this issue in the revised manuscript.

Line 430- You refer to Fig. 16 left panel - which plot are you referring to?

Fig.16- increase font size and label subpanels, refer to labels in text on page 17. Also do this for Fig. 19 and associated text

Figure 16 and 19 have been revised and all the panels have been labeled. The text has been revised to reflect the changes. We hope this make the manuscript easier to follow.

Line 449- You say around 2 dB, but around 2-4 dB would be more accurate

We agree with the referee on this point. Corrected as suggested.

Line 453- Should 2-10 be 2-12 dB?

Yes, it is correct. The change has been done.

Line 463- Is 2.9km correct? On line 445 you say 2.2km.

We have checked the data. On line 445 and 463, it should be 2.5 km. We have corrected this error.

Line 464- Specify that the increase in MVD is only at the beginning of the section.

We have added this information to the revision.

Line 470- I'm sure this will be fixed in later versions, but the figure caption of Fig. 18 is not attached to the figure and appears on a different page.

This issue should be fixed in the final print.

Line 499- I don't understand what you mean by the linear response comment, could you expand on this point please?

What we meant is the linear relationship between the DFRs and the MVD in the small MVD range. When MVD is large (e.g. > 8 mm), the scatter plot presents the "hook feature". We have revised the sentence for clarity.

Line 500- the saturation of DFR Ka/W is discussed in section 4 of Stein et al., 2015. You could perhaps link back to Fig. 2 here, as I don't think you have really made a connection between that figure and your results. Otherwise there is not much need for the figure.

In the revised manuscript, we have superimposed appropriate curves from the modeling work (Fig. 2) to the scatter plots to facilitate the discussion.

Line 503- Figure 21, not 22. Can you just provide a brief sentence explaining how you created these plots of mean MVD? In line 504 you talk about estimating MVD, but you have plotted "mean MVD".

The figure number error has been corrected. We have added a sentence to the figure caption to explain how we computed the parameters.

We have modified some sentence in this paragraph to make the point clear:

Line 505- bulk density decreases as X/Ka decreases and KaW increases- this is not obvious to me. Maybe bulk density decreases as X/Ka increases and KaW increases? Actually page 5757 of Chase et al. (2018) finds that ρe increases as KuKa decreases and KaW increases, as in Kneifel et al. (2015), but I can't see that behaviour from your Fig. 21b. Am I missing something?

As mentioned above, in the revised manuscript, we have improved the presentation of those figures. We hope it illustrates the bulk density rotation feature better.

Line 513- Can you provide references where these DFRs were connected to rimed particles with MVD less than 6mm?

No, we are not aware of. It is from our observations in the study cases.

Line 523/524- Rephrase this sentence beginning "Although..."

In the revision, we have removed that sentence as it is not important for the discussion.

**3 Technical corrections**

We would like to thank the referee for pointing out these errors providing some great suggestions. We have corrected all the errors and revised the text as recommended.

Line 22- there is a double period here, and no period at the end of the abstract. Actually, the abstract seems to be incomplete, unless you mean to say "the analysis shows that there are close relationships between the..."

Line 23- remove s from aggregations

Line 30- Change "its data have shown a great potential also for rain estimation and snowfall in particular" to something like: "its data have shown great potential for rainfall and snowfall estimation"

Line 35- performance is slightly improved

Line 46- multiple radar frequencies, rather than multi radar frequencies

Line 50- instead of "global distribution of the ice-phase precipitation and, therefore, enhancing our knowledge", I would say "global distribution of the ice-phase precipitation, thereby enhancing our knowledge", or "global distribution of the ice-phase precipitation and therefore enhance our knowledge"

Line 53- maybe "capabilities of", instead of "capabilities in"

Line 57- I think the bracket including a sentence along with references separated by commas should be tidied up. One suggestion: "...cloud microphysical data. For example, the OLYMPEX provides 2.2 hours of in-cloud data with Ku-Ka-W radar data and coincident microphysics (Chase et al., 2018; Tridon et al., 2019)."

Line 58- remove such

Line 59- remove comma after low-intensity

Line 64/65- suggested edit: covering a large geographical region and wide range of microphysical conditions

Line 71-75- You are describing what the dataset features, so the points need to relate to this. For example the first point should say "triple-frequency radar data" and the second point should say "data from state-of-the-art in-situ sensors". The second point should end with a semicolon rather than a period. In the third point there is an s at the end of atmospheric.

Line 79- Remove "on"

Line 88/89- Suggest editing this sentence to something like "The large variability in ice crystal properties such as density, size, and shape makes the interpretation of..."

Line 97- Should be "due to non-Rayleigh effects"

Line 102- Remove s from attenuations

Line 109- should be "particle scattering properties", not "particles"

Line 119- "via the discrete dipole approximation". $\mu =$ hasn't come out as a symbol. Need to say what $\mu$ is.

Line 130- provide the highest level

Line 152- Rephrase, suggestion: "Secondly, data from the three radars is..."

Line 153- Change "mapped into" to "mapped onto"

Line 168- change "within 700m from" to "within 700m of".

Line 173- Remove s from snow

Line 195- "For this work, the cloud particle size distribution..."

Line 198- remove space before period

Line 201- should be "determination", remove s from determinations

Line 207 (Table 2)- remove comma after plates

Line 210- "The particle size detection range"

Line 219- Korolev et al., 1998- period should be after al, not after et

Line 226- calculating the total volume within the PSD

Line 227- Do you mean "and aid interpretation of parameters"?

Line 230- the measured PSDs (You are not just using one singular PSD, right?)

Line 232- Heymsfield year, 2004? Put a colon after defined as (rather than a comma)

Line 258- "In the literature, collocating radar and and in situ data is often achieved"

Line 259- finding the nearest airborne radar data points to the in situ measurements

Line 264- on the same platform

Line 265- The temporal sampling rate. Here and on line 269- I don't think there is a section 3.1a and 3.1b?

Line 267- I don't think decimated is the right word, do you mean the radar data is degraded?

Line 271- Remove comma after first

Line 277- in the two directions

Line 287- the equivalent reflectivity factor

Line 288- Rayleigh-Gans

Line 289- Sometimes you write "X-band" and sometimes "X band". Please keep this consistent.

Line 290- scattering effects

Line 303- and densely rimed particles

Line 304- Replace Figure with Fig.

Line 312- (Fig. 9 caption) temperature was in the range

Line 330- Reference to non-existent section 3b, do you mean 3.3.2?

Line 338- (Fig. 11 caption) replace PDFs with pdfs

Line 340- temperatures

Line 344- do you mean "based on"?

Line 344- Fig. 12 shows

Line 351- remove ' from panels

Line 366- change "it is worth to note" to "it is worth noting"

Line 368- rephrase "an increase in dendrites portion". Do you mean "an increase in the proportion of dendritic ice habits"?

Line 389- Change "numbers of concentration are" to "the number concentration is"

Line 390- Change "the section B and C data placement" to something like "the location of data from section B and C in the triple-frequency plane"

Line 394- attributed to

Line 413- Rimed particles and dendrites

Line 419- Move "respectively" to the previous line, after ~ 6 dB.

Line 452- millimetric needs another l. Also, should it be 2mm rather than 3mm?

Line 455- Change 6000 microns to 6mm for consistency with previous units.

Line 457- Change DRF to DFR

Line 459- change boarder to broader. I think you mean to refer to Fig. 18? Change affect to affects

Line 460- Remove "continually" as the increase is for a short period of time and is not continuous over the remainder of the section

Line 461- Change aggregate to aggregates

Line 462- Remove "that"

Line 483- In the abstract and conclusion you say co-located, but in the main text you say collocated.

Line 484- Suggest rephrasing to something like: "provides an unprecedented dataset for studying multi-frequency radar signatures of snow/ice clouds"

Line 485- Perhaps rephrase to something like: "...3.4 hours when the scattering was non-Rayleigh for at least one of the radar frequencies"

Line 486- Rephrase to something like: "illustrated here using data collected from one flight during an Arctic storm..."

Line 487- remove "low" before pristine

Line 488- The dual frequency ratios (DFRs) are as large as 12 dB

Line 488- do you mean "appear to be dominated by" ?

Line 493- Remove s from provides

Line 495- If you take my suggestion for line 488, you don't need to spell out DFRs here

Line 504- Remove s from estimates

Line 510- double period

Line 514- the shape of the PSD also has noticeable effects

Line 528- Change "at reach" to "within reach"

---

## Author Comment (AC4)

Review of the article titled "Coincident In-situ and Triple-Frequency Radar Airborne Observations in the Arctic" by Cuong M. Nguyen et al.

This article shows promising results of triple-frequency radar observations from the Radar Snow Experiment (RadSnowExp). Part of the uniqueness of this article is that both in situ and remotely sensed observations were collected by the same aircraft. Some complications arise when combining these datasets for their analysis, but the authors carefully and thoroughly describe the methodology used for volume matching and range calibration.

The authors studied the relationship between in situ sampled cloud microphysical properties and radar triple-frequency signals particularly, they centered their analysis on the relationships between median volume diameters, effective bulk density and dual frequency ratios. Lastly, the authors suggest a path forward with the possibility for quantitative retrieval of particle size using measured DFR but more in-depth analysis is needed to reach that step.

This article is generally well written, it is interesting, and it will be of interest to the scientific community. Nevertheless, I have a few concerns I suggest be addressed before this article is published. Below are some general and specific comments.

We thank the referee very much for the encouraging remarks.  As some of the results and the dataset presented in this paper are new and unique, the need for clarity and attention to detail is appreciated. As such, we address below the concerns and comments expressed by this reviewer, to the extent we feel feasible.

General Comments:

The authors analyzed one flight (22 November) and divide it into 3 different segments (1948-2000 UTC, 2005-2028 UTC and 2121-2135 UTC) giving a total number of 49 minutes of DFR observations. Please consider this to put into context the overall findings that derive from this dataset that are stated in this article.

We thank the referee for this comment. We have added this information to the revised manuscript.

Figures need significant improvement, both in the actual quality of the figure (suggest improving ppi) and the legibility of axis labels and legends. For figure with more than one panel the different panels should be labeled. This was done for some of the figures but not in all. I suggest a uniform way to address multipanel figures and their caption. Finally, figure captions need to carefully state what is plotted in each panel/figure.

I'm not sure figures 1 and 6 are necessary if the article centers around the Nov 22 flight (in case fig 1 is kept, I made I suggestion below to have all the flights drawn or the domain where all the flights took place). Similarly, I don't think figure 5 is necessary either, no extra analysis is done of how the overlapping sizes were treated or any extra analysis that would

make this plot needed in the article, just stating in the text the size ranges each instrument measures should be sufficient.

We completely agree with the referee on this comment. We have done an intensive modification on the figures. The font size on figure labels and legends are increased. Panels on Fig. 13, 16, and 19 are rearranged and labeled for better readability and presentation. Additional information has been added to the figure captions. We have also removed Fig 1, 5, and 6 as they are not necessary for the paper as suggested. With this major modification we believe the revised manuscript would better facilitate the reading.

Specific Comments:

Lines 14-16: Consider stating the amount of data that was used to reach the conclusions stated in the article and not the overall flight hours of RadSnowExp.

We have revised the sentence as suggested.

Line 18: I'm not entirely sure this article showed how to accurately derive the level of riming from the DFR plane, consider rewriting this sentence.

We have rewritten this sentence.

Line 26: Add the definition of GPM-DPR

We have added the definition of GPM-DPR

Lines 29-31: Reference needed.

We have added Haynes et al., 2009; Hiley et al., 2011; and Matrosov et al., 2008.

Line 34: Suggest modifying this sentence as "The GPM Core Observatory carries…" or similar.

We have modified the sentence as suggested.

Lines 39-41: Reference needed.

The reference has been added.

Lines 63-65: Figure 1 does not support this statement, a figure showing all the flight tracks, or the experiment domain would be more useful.

As it is pointed out by the referee and others, we have remove Fig. 1 because it is not necessary for the paper. We have revised the text to reflect the change.

Line 71: Consider replacing 'uniquely' by 'unique'

We have corrected this error.

Line 71: Figure 3 should be Figure 2, please reorder figures.

In the revised manuscript, we have removed Fig. 1, 5 and 6 and changed the text accordingly.

Line 110: Should be figure 3 when re-ordered.

This error has been fixed.

Line 110: consider replacing 'points' by 'retrievals' or similar.

We have replaced it with "data points".

Lines 149-154: The second bullet point is not clear. What do you mean by 'three radar data'? How was it mapped into a common range axis? How 'reasonable' is the homogeneity assumption? Was any sensitivity analysis done to evaluate it? A schematic of all the smoothing methods could be valuable to assess the actual volumes that are being compared.

We thank the referee for this comment. We have corrected the phrase "three radar data" to "data from the three radars" and also revised the text in this bullet for clarity. The common range axis has origin at the aircraft location and a 35 m grid. The data mapping is done using a standard interpolation method. Because the common range spacing (35 m) is greater than the sampling resolutions of the three radars the standard interpolation method works well. We found that the assumption of homogeneity of clouds is only needed within the largest radar volume and within 50 m around the aircraft. This assumption may not be met at large range and/or near the boundary of cloud/precipitation. In our case, the radar data used in the analysis are at 245 m from the aircraft. At this range, the radar volumes are small (less than 20 m in antenna beam width for a 4.2 deg beam) so that this assumption could be made.

We were aware that beam matching is a critical aspect for triple frequency radar analysis and considered the best approach to process the data. The impact of mismatch beams/non-uniform beam filling for the triple frequency analysis need to be addressed but it is beyond the scope of this paper. In the revision, we have removed the word "reasonable" to avoid the confusion.

Lines 175-179: I consider this to be an important factor in the data analyzed here that could grant the inclusion of a figure showing these values to support this statement.

We completely agree with the referee on this point. In the revised manuscript, we have added the scatter plots of cross-calibrated W, Ka and X reflectivities at 245 m from the nadir antennas in the 22 Nov flight for the region of median size < 300 um.

Line 180: Figures should be numbered in sequential order they are referenced in the text. Consider reorganize the figures or the text.

We have modified the text to address the figure order issue.

Line 181: Were there more than 1 flight on 22 Nov? How much data does this represent? i.e.: How many data points were used to reach this conclusion?

There was only on flight on 22 Nov. We used 510 data points where the aircraft sampled regions of small ice crystal (median size <300 um) to verify the cross-calibration between the frequencies. In the revised manuscript, we have included scatter plots of the reflectivities in the regions of small ice particles to illustrate the conclusion.

Lines 193-194: Consider adding sizes to contextualize 'small cloud droplets' and 'large precipitation hydrometeors'.

We have added sizes as suggested.

Line 197: Consider replacing 'or' by 'and'.

We have revised the text as suggested.

Line 206: Is the difference between all the 9 groups and the subset of ice habits only the inclusion or not of the Drops and Artifacts category? How are small particles treated?

The subset of ice only includes pristine, dendrites, rimed dendrites, rimed particles, aggregates and other ice particles. Small particles are not in the subset of ice habits.

Lines 212-213: It would be good to state the uncertainty values that are within the range presented by Baumgardner et al. (2017).

We computed the uncertainty values in sizing and concentrations from a wind tunnel calibration dataset. However, the uncertainty values in flight environment could be different.

Lines 219-224: This paragraph is confusing, if the wiring had little effect on the estimated water content, then what was the factor that made the accuracy drop from 0.002 g/m3 to 0.05 g/m3? Was this value taken as a constant value regardless of the size of the hydrometeors sampled? How was the estimation of the accuracy of the Nevzorov probe done?

We have revised the paragraph for clarity.

First, we would like to highlight that 0.002 g/m3 is the sensitivity reported by (Abel et al., 2014) for this commercial sensor, while 0.05 g/m3 is our estimated accuracy due to uncertainty in the dry term calculation. It is reasonable to expect that in the presented airborne measurement, LWC and IWC may be underestimated, especially in segments with higher MVDs. Unlike in wind tunnel studies, the quantification of this bias is challenging due to the complexity of verification of IWC measurements in naturally occurring ice particle population. In other flight campaign (e.g. Faber et al. 2018), similar uncertainty values were found. We included this clarification in the text.

Line 226: If the minimum of 50 µm was used as lower bound then FCDP was not used in this analysis?

The threshold 50 µm is only used in the calculation of effective bulk density. For MVD, number of concentration (Nt) and other qualitative assessments, we use the full spectrum of the PSD.

Line 229: Consider adding "(The definition of) several bulk…"

We thank the referee for this comment. We have revised the text as suggested.

Line 232: Add year to Heymsfield et al.

We have added the year to this reference.

Lines 242-248: I think this paragraph and figure 6 could be removed from the article. Just adding a sentence that states that 22Nov will be analyzed and why should be sufficient.

We agree with the referee. In the revised manuscript, we have removed Fig. 6 and amended the text accordingly.

Line 258: Consider adding "In (current or past) literature…" and adding several references of this literature.

We have added some references to this sentence in the revision.

Lines 266-267: Convair average ground speed is 100m/s, this makes the volume for the in situ sampling of 200-500 m. How is this mismatch between the radar and in situ volumes handled?

The in situ probe sampling volume (Baumgardner et al., 2017) is much smaller compared to the radar volume. We believe there is no processing technique to match volumes from the in situ probes and the radars. In our processing, the radar data is decimated to match with the temporal resolution of the in situ data.

Lines 271-272: I suggest analyzing if reflectivity shows a different signal comparing Nadir vs Zenith samples and then compare DFR.

In this flight, the aircraft stayed in inhomogeneous cloud layers most of the time. Figure 11 (pdf of the X-band reflectivity at the nearest range above and below the aircraft) shows the difference between the nadir and zenith data with higher reflectivities typically occurring below the aircraft. This is the reason we decided just to compare DFR directly.

Section 3.3.1: This paragraph and associated plot is confusing, the almost 500 m difference between the observations at nadir and zenith show that the vertical structure of the cloud has a large impact on DFR. This is particularly clear when comparing DFR X/Ka, where the Zenith ratio has an almost constant value regardless of the values measured at Nadir. Is

there a reason for this? As mentioned before, joint distributions of reflectivity would be better to analyze this 500 m effect.

Unfortunately, as we have discussed in section 3.1.2, the closest useable triple frequency radar data are 245m below and above the aircraft. Hence, there is almost 500m between the DFR observations. The example (Fig. 7) shows that in mixed phased clouds the DFR could vary in a wide range in a short distance (<500 m). In this section, we want to emphasize that due to possibly large variability in DFR observations, matching in situ and radar data is critical. In the OLYMPEX airborne campaign (Chase et al., 2018) when they flew two aircrafts, the radar collected within 10 minute temporally and 1 km spatially of the in situ are regarded as collocated. Since we wanted to focus on the DFR variability, we opted to show the joint distributions of DFRs instead of reflectivies.

Line 288: Add 's' to Gans.

The typo has been corrected.

Line 294: Figures should be numbered in sequential order.

We have changed the text to address this issue.

Lines 292-294: I'd consider regions where both the Nadir and the Zenith correlation coefficient are good. This will hint at a more homogeneous cloud in the vertical and thus a more reliable comparison between what is sample in situ and at a 245 m difference in height. For example, the 4[th] region marked in figure 10 shows a big difference between Nadir and Zenith (based on correlation coefficient) this is most probably hinting that the part of the cloud sampled has a notable vertical structure so, I'm not sure it is a good case to analyze DFR because this gives an extra reason for the differences between the DFRs.

We agree with the referee that it would be best to select regions where both the nadir and the zenith correlation coefficient are good. However, because in most part of this flight we sampled inhomogeneous clouds, this condition is hard to be met. In the first and second flight segments, the nadir correlation coefficient is good (≥0.6) and the zenith correlation coefficient is also good for most part of the segments. In the last segment the nadir correlation coefficient is high whist the zenith correlation coefficient is greater than 0.5 most of the time so we still believe it is a good case for DFR analysis. We have attached a figure showing the histogram of the nadir correlation coefficient. The histogram show a main mode with correlation coefficient greater than 0.6. Hence, we selected a threshold of 0.6 for the decision of a good match between the in situ measurements and the radar data.

[Figure]

Lines 296-305: How long was this flight? How many samples were analyzed?

The flight was 3.5 hour and we analyzed 49 minutes (or 588 samples) of triple-frequency observations in the three study cases. We have added this information the manuscript.

Line 333: Replace 'similarity measurements' by 'correlation coefficient filter' or cc threshold or similar.

We have revised the text as suggested.

Line 339: How where the different sections within segment 1, 2 and 4 defined? Was this breakdown into sections defined by the aircraft sampling pattern? Because figure 13 shows that there are different processes occurring in these different sections, for example, section D shows clearly different behaviors in DFR and CPI particle fraction near the beginning of the section when compared to the end of the section.

The decision for the breakdown into sections within the segments is based on the differences in the cloud processes, the observed DFRs, and the bulk measurements which are all related. We found that by naming different sections, it is easier to describe the cases and to present the results. It does not depend on the aircraft sampling pattern.

Lines 340-346: It'd be best if this description of the first segment has figure 13 as reference, it'd help contextualize the differences in the different segments.

We agree with the referee on this point. In the revised manuscript, we have combined figure 12 and 13 and amended the text accordingly.

Lines 345-346: The bimodality in the PSD distributions it difficult to see for all sections, especially the referred maxima at 1 mm. Please clarify.

In the revision, we have changed the text to "The mass distributions are generally bi-modal with two ice modes around 30 µm and 1 mm"

Line 351: IWC should be TWC or is the legend in the figure incorrect?

It should be TWC. The error is now corrected.

Line 357: Why/How was the aircraft at 6 km height? I assumed the black line in Figure 10a is the aircraft path, so in sections b-d shouldn't the aircraft be at around 2.4 km height? It would be extremely beneficial to have the different sections shown in figure 10a.

We thank the referee for pointing out this error. It should read "at about 2 km". We agree with the referee having the different sections shown in figure 10a is helpful. We tried to implement this idea but the figure is getting too busy; especially when we also added modelling lines to those figures. We might try different idea to address this concern in the final version of the manuscript.

Line 364: Consider rewriting 'remarkably mirrors'

We have changed it to "resembles".

Line 365: Similar to the comment before, from Figure 10a after the first few minutes of the first segment (first few minutes of section a that the aircraft descended) the aircraft seems to be flying at a constant altitude of ~2.4 km, is this not the case? Aren't sections A-D correspond to segment 1 that is shown by the first box in figure 10a?

We apologise for this confusion. The text was messed up from another section. It is correct that the aircraft stayed at a constant altitude (~2.4 km) during sections B-D. We have corrected this mistake.

Lines 372-373: This sentence that the fraction of dendrites and rimed particles drops to its lowest level in section D can be misleading, this is the case near the beginning of the section, but by the end of the section this is clearly not the case. Consider rephrasing to avoid confusion.

We have changed the text to " … rimed particles drops to its lowest level at the first half of the section when the highest DFR X/Ka occurs" for clarify.

Lines 390-391: Would it be possible to add to figure 14 the line representing graupel particles using discrete dipole approximation? It could be helpful to add the relevant curves from figure 2.

We agree with the referee. The relevant curves from Fig. 2 have been added to Fig. 14, 17, and 20.

Line 405: Consider ordering figures in sequential order that are mentioned in the article (Figure 15 should come before Figure 16). Also, how where these different sections defined?

In the revised manuscript, we have merged Fig. 15 into Fig. 16. We believe it would help the readers follow the analysis easier.

Lines 409-412: From figure 16 top right panel it seems like the fraction of dendrites is higher in C than in B?

We thank the referee for pointing out this error. The sentence has been corrected "In section B, the fraction of rimed particles is the highest."

Line 419: MDV does not reach 6 mm in section A, please clarify.

We have amended the text to correct this error. It now reads "…and DFR X/Ka at times reached the same level as DFR Ka/W at around 8 dB when MVD exceeds 4 mm."

Lines 420-421: ZDR was not mentioned before in the article, was there not any ZDR signature in the previous segment?

We thank the referee for this comment. We have added a sentence on the ZDR observation for the first segment.

Line 421-423: Consider rewriting this sentence, variables do not mimic other variables. Also, MDV is not > 8 mm for all the times that DFR is ~ 10 dB, this occurs just for the second maxima in DFR in section B.

We have changed the text and also corrected for the value of MVD " … and MVD is greater than 6 mm".

Line 423: consider adding a time series of ZDR to figure 16.

The time series of ZDR was already plotted in the second panel of the CPI plots in Fig. 16.

Lines 426-427: What do you mean by 'fluctuations in the DFRs'? Also, what datapoints correspond to section A, B or C is not clear from the plot, consider making the markers edge a different color linked with each section. Also, I suggest adjusting the limits of the plot to better fit the data plotted this will make the differences in the markers size and colors clearer.

We have rephrased the text 'fluctuations in the DFRs' in the revision. Regarding the scatter plot, we would like to keep the axis limits consistent between all the study cases and the modelling work (Fig. 2). In the revised manuscript, we have used different colors for the edge the dots to help identify sections A-C easier.

Lines 429-435: These few sentences are confusing consider rewriting. The hook feature is present in data from section C not B. Also, remove parenthesis for the references inside the larger parenthesis, like in the Petty and Huang, 2010 reference.

We have rewritten the paragraph for clarity. Data from all the section (A-C) present a hook feature and the turning point occurs at section. We have also corrected for the redundant parenthesises in the references.

Line 444: Why was this segment chose to be analyzed in depth? Please consider the previous comment regarding the difference between observations at Nadir and Zenith with respect to the vertical structure of the cloud and how different processes can be playing a role at different heights of the clouds that could give difference in DFRs that are not exclusively related to the factors analyzed here. This variability in height can clearly be seen in figure 10a where different microphysics might be acting between the lowest trusted range sampled by the radar and the in situ observations sampled 245 m below.

In this segment, we sampled a region of high concentration of supercooled drop with heavy ice accretion and then a region of milimetric rimed particles and large aggregates. The shape of the hook feature in Fig 19 is different from the previous study cases. In section C of this segment, DFR Ka/W goes up to 10-12 dB at MVD ~ 6 mm similar to the previous segment but DFR X/Ka is much lower at 5-7 dB. This is a feature which, we believe, is worth to be analyzed in depth.  We agree with the referee that in this segment, there is difference in the nadir and zenith observations but as mentioned in a response before, we decided to use this segment because the nadir data is identified as good match based on the correlation.

Lines 484-485: This sentence is misleading as a summary and discussion part of this article where 1 day was analyzed and 3 segments of that flight that resulted in 49 minutes of DFR observations.

We have added "The whole RadSnowExp dataset …" to avoid the confusion with the flight we analyzed in this paper. The next sentence should make it clear to the readers "The potential of this dataset is illustrated here using one flight data during an Arctic storm that covers a wide range of snow habits from …"

Lines 488-490: This sentence is confusing, please consider rewriting it.

We have rephrased this sentence.

Lines 496-499: Add 'for the flight we analyzed' or similar phrase here as the results described could be case dependent.

We have added the text as suggested.

Lines 503-504: Figure 22 should be figure 21?

Yes. Correction has been made.

Line 503: I'm not sure I understand how figure 22 (or 21) is a first attempt for quantitative retrieval of particle size using measured DFR Ka/W and DFR X/Ka. There are a lot mean MVD and density values that are linked with different DFRs values. This looks more like a qualitative analysis.

We have revised the sentence for clarity. It now reads "Reversely, the strong connection between the particle size and the triple-frequency radar signature suggests that the data could be directly used to produce look-up-tables for mapping measurements in the (DFR Ka/W, DFR X/Ka) space into microphysical properties like median volume diameter and effective bulk density with associated uncertainties.". We also added standard deviation plot of the parameters to Fig. 21.

Line 505: Figure 21b shows that equivalent density increases with decreasing both DFR

Actually that sentence is not needed for the discussion in this paragraph. In the revision, we have removed it. The discussion of the $\rho_e$ rotation feature is better illustrated in study case 3.

Line 510: Remove extra '.'

This error has been corrected.

Lines 515-517: Change 'demonstrated' by 'shown', otherwise, this is too strong of a statement based in the case analyzed in this article.

Correction has been made.

Lines 523-524: This sentence is not clear, consider rewriting it for clarity.

The sentence is not necessary for the discussion and has been removed.

Fig. 1: please improve figure 1a, it is very difficult to visualize the location of the flights, what is the color in the track representing? Not sure how convoluted a figure showing the path of all the flights done in RadSnowExp, but it'll sure be useful to see such figure.

In the revised manuscript, we have removed Fig. 1, 5 and 6 because they are not necessary for the paper.

Fig. 4: Consider center this figure around the heights that are referenced in the article. For example, if the ground is not referenced why extend the y-axis all the way 5 km?

We have modified this figure to include a "zoom in" section at the close range region. The ground is shown to illustrate the three radar profiles are well aligned (section 3.1.1).

Fig. 7: Please add in the figure caption what is shown in the figure (what is the dashed line and the error? bars).

The dashed line is the simple 1:1 line and the black line presents the data means of the DFRs with error bars of one standard deviation. We have added this information to the caption of Fig. 7.

Fig. 8: Consider adding lat and lon to the plots and make the maps larger the synoptic map is not very useful in the context given for people not familiarize with the area the flights took place to easily link all 3 maps.

We have added the lat and lon lines to the Fig. 8b.

Fig. 9: What is the F9 legend? What do you mean by 'the ground to air temperature' ground temperature is ~ -23C and why is this important information to have?

Fig. 9 has been revised. The temperature is now plotted as a function of altitude. Proper legend has been included.

Fig. 10: Improve figure caption to add what are the red, blue and black lines in panel (b) and also improve the legibility of the legend in panels b-d. Also, please consider not using similarity measurement and just use correlation coefficient.

We have increased the vertical size of the figure and the font size of figure 10 to improve the readability. The caption has been revised as suggested.

Fig. 12: Please correct what figure is referencing the five sections. Also, I don't see a reference in the article that requires panel b to be part of the figure.

In the revision, we merged Fig, 12 and 13 into one figure to facilitate the reading. The Fig. 12b would be referred in line 346 of the original manuscript.

Fig. 13: Please label each panel of the figure for an easier read and to improve the connection between the text and each plot in the figure.

We have modified Fig. 13 significantly as suggested by the referee and other reviewers.

Fig 14: Consider modifying the colorscheme of the scatter plot to improve the readability of the figure (pinkish colors represents both low and high values).

The figure's colormap and display scale have been changed for better readability.

Fig. 15b: I don't see a mention to this panel in the article.

It is mentioned in line 429-431 in the manuscript. In the revise manuscript, we have combined Fig. 15 and Fig. 16.

Fig. 16: Similar to Figure 13 please label each panel.

We have added labels to all the panels.

Fig. 21: A joint distribution with number of samples would give reference to the mean MVD and density.

We have added number of samples used to compute the mean MVD and density.

Fig. 22: Add what the black line and vertical black lines are in the figure caption.

We have added the description of the black line to the caption of Fig. 22.